



# Surface temperatures and their influence on the permafrost thermal regime in high Arctic rock walls on Svalbard

Juditha Undine Schmidt[1], Bernd Etzelmüller[1], Thomas Vikhamar Schuler[1], Florence Magnin[3], Julia Boike[4,5], Moritz Langer[4,5], Sebastian Westermann[1,2]

[1]Department of Geosciences, University of Oslo, Oslo, 0316, Norway
[2]Center for Biogeochemistry in the Anthropocene, University of Oslo, Oslo, 0316, Norway
[3]EDYTEM Lab, Université Savoie Mont Blanc, CNRS, Le Bourget-du-Lac cedex, 73376, France
[4]Alfred Wegener Institute (AWI), Helmholtz Centre for Polar and Marine Research, Potsdam, 14473, Germany
[5]Department of Geography, Humboldt-Universität zu Berlin, Berlin, 12489, Germany

*Correspondence to*: Juditha U. Schmidt (juditha.schmidt@geo.uio.no)

**Abstract.** Permafrost degradation in steep rock walls and associated slope destabilization have been studied increasingly in recent years. While most studies focus on mountainous and sub-Arctic regions, the occurring thermo-mechanical processes play an important role also in the high Arctic. A more precise understanding is required to assess the risk of natural hazards enhanced by permafrost warming in high Arctic rock walls.

This study presents rock surface temperature measurements of coastal and non-coastal rock walls in a high Arctic setting on Svalbard. We applied the surface energy balance model CryoGrid 3 for evaluation, including adjusted radiative forcing to account for vertical rock walls.

Our measurements and model results show that rock surface temperatures at coastal cliffs are up to 1.5 °C higher than non-coastal rock walls when the fjord is ice-free in the winter season, resulting from additional energy input due to higher air temperatures at the coast and radiative warming by relatively warm seawater. An ice layer on the fjord counteracts this effect, leading to similar rock surface temperatures as in non-coastal settings. Our results include a simulated surface energy balance with short-wave radiation as the dominant energy source during spring and summer, and long-wave radiation being the main energy loss. While sensible heat fluxes can both warm and cool the surface, latent heat fluxes are mostly insignificant. Simulations for future climate conditions result in a warming of rock surface temperatures and a deepening of active layer thickness for both coastal and non-coastal rock walls.

Our field data present a unique data set of rock surface temperatures in steep high Arctic rock walls, while our model can contribute towards the understanding of factors influencing coastal and non-coastal settings and the associated surface energy balance.



## 1 Introduction

As a response to a warming climate, degradation of permafrost in steep rock walls can contribute to slope destabilization (Gruber and Haeberli, 2007; Krautblatter et al., 2013). Increased frequencies of slope failures have been observed in recent years (Fischer et al., 2012; Gruber et al., 2004; Ravanel et al., 2010, 2017). These natural hazards can damage infrastructure and cause casualties in downslope regions (Harris et al., 2001, 2009). Permafrost in rock walls has been studied in mountainous

regions (Allen et al., 2009; Krautblatter et al., 2010; Magnin et al., 2015; Noetzli and Gruber, 2009) as well as in sub-arctic areas (Blikra and Christiansen, 2014; Lewkowicz et al., 2012; Magnin et al., 2019). However, permafrost dynamics in steep rock walls in the high Arctic are poorly understood. In this study, we will focus on rock surface temperatures in steep coastal and non-coastal cliffs at a high Arctic site near Ny-Ålesund, Svalbard (Fig. 1).

Svalbard is located in the northern part of the warm North-Atlantic current and therefore, it is very sensitive to atmospheric and oceanic changes (Walczowski and Piechura, 2011). Increasing air temperatures are observed for more than a century (Nordli et al., 2020). Climate models predict an increase in precipitation and a warming of air temperature with the most pronounced air temperature change in winter season (Hanssen-Bauer et al., 2019; Isaksen et al., 2016). The climatic changes are also apparent in permafrost temperatures on Svalbard as observed in boreholes over the last decades (Boike et al., 2018;

Christiansen et al., 2010; Isaksen et al., 2007). Simulated thermal conditions in Svalbard show an increase of ground temperatures and indicate a significant warming and an increase in active layer thickness over the 21$^{st}$ century (Etzelmüller et al., 2011). Besides large-scale climate changes, local conditions can play an important role in the surface energy budget, resulting in an amplification or dampening of the large-scale signal (Westermann et al., 2009). Besides sensible and latent heat fluxes, short-wave and long-wave radiation are crucial factors as they have a strong impact on the energy transfer processes

from the atmosphere to the ground, effectively modulated in presence of insulating snow cover. The terrain exposure induces significant spatial variability of short-wave radiation that should be considered when modelling thermal conditions in steep rock walls (Fiddes and Gruber, 2014; Magnin et al., 2015).

Besides down-welling radiation, long-wave radiation emitted by water bodies as well as reflected short-wave radiation on

snow and ice can influence the rock surface temperature. Therefore, sea ice coverage plays an important role for the surface energy balance of coastal cliffs. According to observations since 1997, Kongsfjorden was characterized by sea ice cover during winter season (Gerland and Hall, 2006). Since 2006, the sea ice extent has been reduced significantly and the ice thickness and snow cover on ice have become thinner (Johansson et al., 2020). This could also affect coastal erosion as sea ice and development of an ice foot protect the cliffs by absorbing ocean wave energy and control the removal of weathered material

from the base of the cliff (Ødegård and Sollid, 1993). With shorter or absent fast ice periods, coastal cliffs are exposed to waves and tides for longer durations. Climate models predict a further reduction of sea ice cover in the western fjords of Svalbard (Hanssen-Bauer et al., 2019). Thus, thermal models for steep rock walls have to consider the influence of aspect and

slope angle on radiative forcing as well as additional heat sources like open seawater and reflection of short-wave radiation on sea ice.


In this study, we applied a full energy balance model to evaluate the role of the different radiative forcing elements on the thermal regime in steep slopes at a high Arctic site. In doing so, we extended the parametrization of radiative forcing in the thermal model CryoGrid 3 to account for effects governing steep rock walls, and validated the model with measured rock wall temperatures in the study area. Our objectives were to analyze the effect of coastal and non-coastal settings (i) on rock surface

temperatures of vertical rock walls, (ii) the surface energy balance throughout the seasons and (iii) to estimate future developments of the thermal regime until 2100 for these settings.

## 2 Study site

The observation site is situated near the village of Ny-Ålesund, Kongsfjorden, located at the west coast of Spitsbergen. We measured rock surface temperatures in steep coastal and non-coastal rock walls (Fig. 1). Carbonate rocks of Permian to

Carboniferous age with an apparent joint system are the dominant bedrocks (Fig. 2). The surrounding of the study area is a strandflat and characterized by tundra vegetation, while the surface sediments are dominated by fine to medium-grained glacial and marine deposits (Hop and Wiencke, 2019; Westermann et al., 2009).

Long-term records of climatological parameters are evidence of ongoing changes in the Arctic climate system with an increase

of mean annual temperature by $+1.3 \pm 0.7°C$ per decade and a rise during winter months by $+3.1 \pm 2.6°C$ per decade. The winter warming is linked to a change in downward long-wave radiation of $+15.6 \pm 11.6$ $Wm^{-2}$ per decade (Maturilli et al., 2015), while the net short-wave radiation is altered by a decrease in reflection caused by reduced snow cover duration (Hop and Wiencke, 2019).

The main surface wind direction is along the axis of Kongsfjorden from the inland to the coast throughout all seasons. The mountains cause complex wind fields (Maturilli and Kayser, 2017) and a south-easterly wind flow occurs as a result of channeled winds from the Kongsvegen glacier (Beine et al., 2001). Measured mean annual precipitation in Ny-Ålesund in the period 1971-2000 was 409 mm (Hanssen-Bauer et al., 2019). It can occur as both rain and snow throughout the year, but the snow-free season is typically from June to October (Hop and Wiencke, 2019).



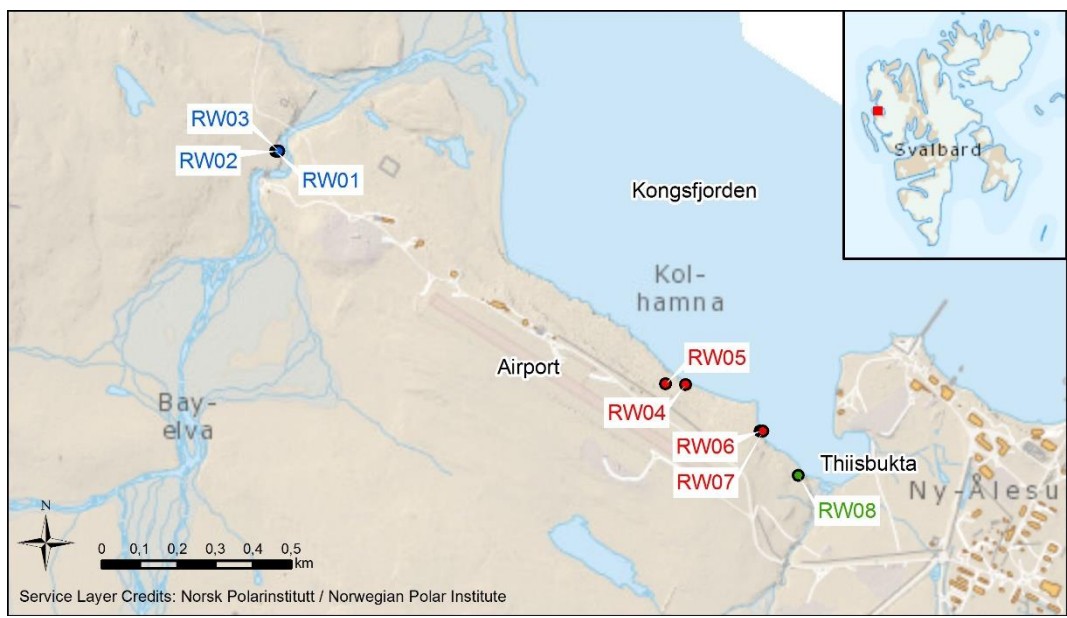

**Figure 1:** Locations of installed logger in the canyon (blue labels: RW01 to RW03), at the coastal cliffs (red labels: RW04 to RW07) and in the bay Thiisbukta (green label: RW08). Source: NP_Basiskart_Svalbard_WMTS_25833 / FKB_Svalbard_WMTS_25833, ETRS 89 UTM 33 © Norsk Polarinstitutt (npolar.no).

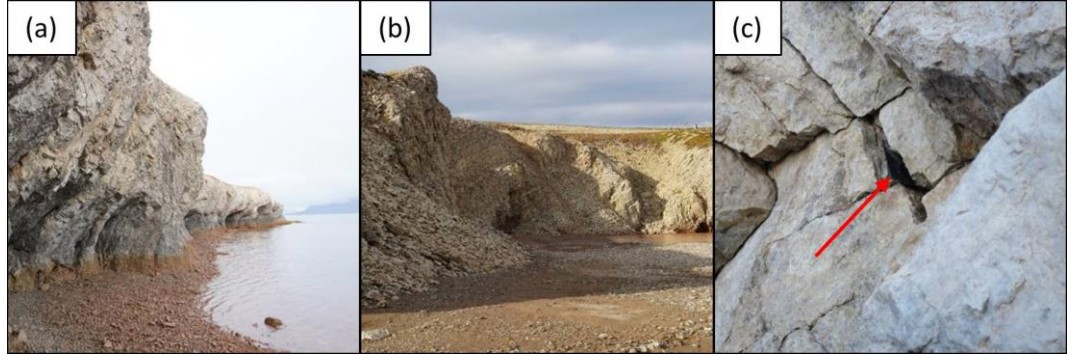

**Figure 2:** Locations of rock wall loggers used in this study: (a) coastal cliffs at the open fjord next to Ny-Ålesund airport (tidal zone visible in bottom); (b) non-coastal rock walls in the canyon of Bayelva; (c) close-up of a rock wall logger location: marking tape is visible, while the logger is located about 5 cm inside the crack in thermal contact with the rock.

## 3 Methods

### 3.1 Surface rock temperature monitoring

In this study, we used eight iButton (© Maxim) temperature loggers with an accuracy of +- 0.5 °C for this study (Table 1). They were installed during summers and springs of 2016 and 2017 in different locations, labelled with RW01 to RW08 (Fig. 1) near Ny-Ålesund in Svalbard and were placed on north- to east-facing rock walls. They represented three different settings: (i)



non-coastal rock walls in the canyon of Bayelva (three locations), (ii) coastal cliffs at the open fjord next to Ny-Ålesund airport

(four locations) and (iii) a coastal cliff in the bay of Thiisbukta (one location). The settings allowed the analysis of permafrost

temperatures in non-coastal rock walls and coastal cliffs affected by seawater (Fig. 2).

**Table 1: Settings of surface temperature loggers used in this study at eight different locations RW01–RW08 in the surroundings of Ny-Ålesund, Svalbard.**

| Location | Site | Time period | Aspect |
| --- | --- | --- | --- |
| RW01 | Non-coastal | 27.08.2016 – 27.08.2020 | NE |
| RW02 | Non-coastal | 27.08.2016 – 27.08.2020 | NE |
| RW03 | Non-coastal | 27.08.2016 – 27.08.2020 | NE |
| RW04 | Open fjord | 27.08.2017 – 27.08.2020 | NE |
| RW05 | Open fjord | 31.08.2016 – 27.08.2020 | N |
| RW06 | Open fjord | 12.05.2017 – 27.08.2020 | ENE |
| RW07 | Open fjord | 12.05.2017 – 27.08.2020 | NE |
| RW08 | Bay | 31.08.2016 – 27.08.2020 | NE |


## 3.2 Model description

We adapted the CryoGrid 3 ground thermal model (Westermann et al., 2016), originally designed for horizontal surfaces, to

account for conditions in steep rock walls (Magnin et al., 2017). CryoGrid 3 calculates rock temperatures by solving the heat

equation, uses the surface energy balance as an upper boundary condition, and considers latent heat effects depending on water

content of the substrate. The heat transfer to the ground is calculated by heat conduction. The surface energy balance is derived

from time series of air temperature, specific humidity, wind speed at a known height above the ground, incoming short-wave

and long-wave radiation, air pressure and rates of snowfall and rainfall (Westermann et al., 2016). In the standard version

designed for horizontal surfaces, turbulent fluxes between the surface and the atmosphere are controlled by vertically moving

air parcel as defined in the Monin-Obukhov similarity theory (Monin and Obukhov, 1954). As a consequence, movement of

air parcels at a vertical wall would be parallel to the surface rather than perpendicular. Therefore, we assumed in all model

calculations, that the near-surface wind profile follow a neutral atmospheric stratification (Magnin et al., 2017).

Besides the analysis of rock surface temperatures (RST), we used CryoGrid 3 to determine the active layer thickness (ALT).

We applied a small grid spacing in the upper layers (0.1 m between 0 m and 1 m depth) and gradually increased the grid

spacing to the lower layers of the model (10 m between 50 m and 100 m depth) to account for detailed ground temperature

calculations in the active layer near the surface.

We define the surface as the interface between the atmosphere and the rock wall. Fluxes, which transport energy away from

the surface have a negative sign, while fluxes, which transport energy towards the surface are denoted positive.



### 3.3 Preprocessing

As the energy input of short-wave and long-wave radiation is depending on varying aspects and slope angles of the rock walls,
we modified the model to account for the different physical settings of the logger locations. We calculate incoming short-wave radiation as the sum of direct, diffuse and reflected short-wave radiation, while incoming long-wave radiation includes atmospheric long-wave radiation as well as heat emission of the close environment.

We divided short-wave radiation into direct and diffuse components (Fiddes and Gruber, 2014). It required the determination of the atmospheric clearness index $k_t$, the ratio between solar radiation arriving at the surface $S_{in}$ and the radiation at the top
of the atmosphere $S_{TOA}$:

$$k_t = \frac{S_{in}}{S_{TOA}}. \tag{1}$$

The fraction of diffuse short-wave radiation $k_d$ was computed based on the clearness index $k_t$

$$k_d = 0.952 - 1.041e^{-\exp(2.300-4.702*k_t)} \tag{2}$$

and taking into account the sky view factor SVF. As we applied the model to vertical rock walls, we assumed a SVF of 0.5 for
all locations (Kastendeuch, 2013). Therefore, the amount of diffuse short-wave radiation $S_{diff}$ can be expressed as

$$S_{diff} = SVF * k_d * S_{in}. \tag{3}$$

Consequently, the amount of direct short-wave radiation $S_{dir}$ is the remaining fraction $(1 - k_d) * S_{in}$. After we determined the azimuth $\alpha_{sun}$ and elevation $\beta_{sun}$ of the sun for every time step depending on latitude, longitude and altitude of each location, we projected direct short-wave radiation $S_{dir}$ on inclined slopes (Appendix A).


Besides direct and diffuse short-wave radiation, we implemented reflected short-wave radiation in the model to account for diffuse reflection on ice and snow surfaces as well as on snow-free terrain. Assuming Lambertian reflectance, we derived reflected short-wave radiation $S_{ref}$, taking into account the albedo of the surface $\alpha$ and the sky view factor SVF:

$$S_{ref} = S_{in} * \alpha * (1 - SVF). \tag{4}$$

We used the sum of diffuse, direct and reflected short-wave radiation as a driving variable for the model CryoGrid 3 on vertical rock walls.

Moreover, we modified long-wave radiation by using the implemented sky view factor SVF. For simplicity, we assumed an SVF of 0.5 for all locations, so 50 % of the long-wave radiation is given by the forcing data $L_{in\_forc}$, representing the
atmospheric long-wave radiation, while the rest is derived from the ambient temperature $T_{amb}$ applying the Stefan-Boltzman law with Stefan-Boltzman constant $\sigma$ (Fiddes and Gruber, 2014):

$$L_{in} = SVF * L_{in_{forc}} + (1 - SVF) * \sigma * (T_{amb} + 273.15)^4. \tag{5}$$





With this approach, we assumed that incoming long-wave radiation is isotropic. The ambient temperature $T_{amb}$ was given by either the air temperature, or the sea surface temperature in case the logger is located directly above the sea. If the seawater

was covered by ice and could not emit any heat, we used air temperature for deriving the long-wave radiation.

Apart from the modification of incoming short-wave and long-wave radiation, we included the water balance in the model by implementing a water bucket approach. Due to the vertical alignment of the rock walls and its consistency of hard bedrock, precipitation does not infiltrate into the material and evaporation of moisture at the rock surface dominated the latent heat flux.

Therefore, the latent heat flux had only minor influence on the total surface energy balance and a simplistic water bucket approach was sufficient for the required model setup (Appendix B).

We did not consider snow cover in the model, which, according to field observations in April and May, was adequate for the majority of measurement sites. An exception is displayed in Fig. 3, showing the damped signal of RW01 due to snow cover.

The data of these time periods were analyzed but excluded from the comparison with the model.

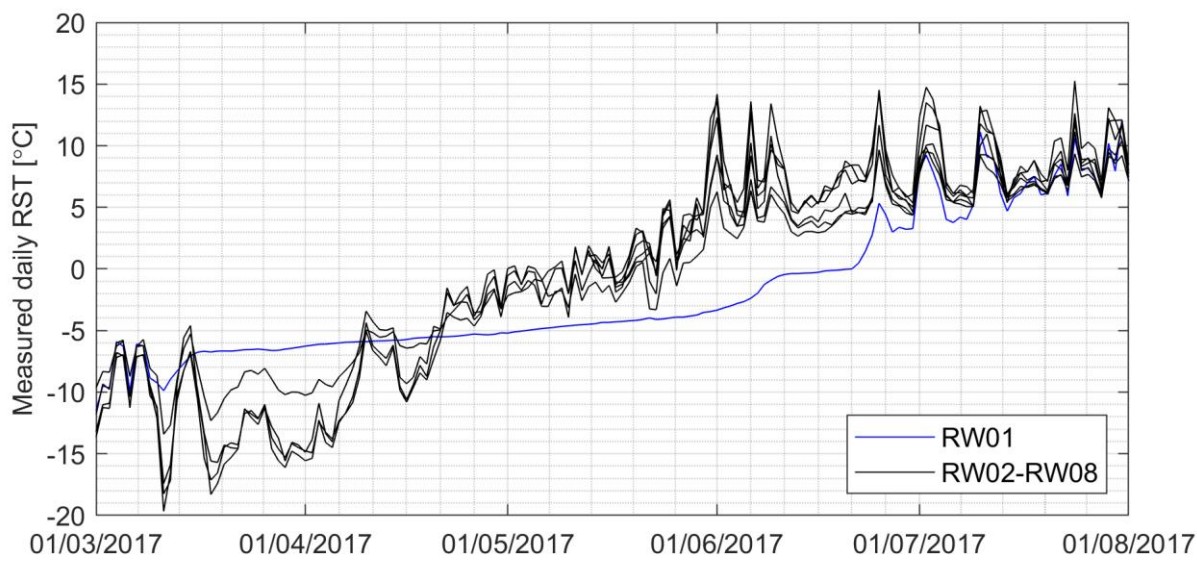

**Figure 3: Mean measured daily RST in winter 2017. RW01 shows a temporarily damped signal due to snow cover.**

### 3.4 Model parameters and forcing data

The study focused on rock surface temperatures, thus simplified subsurface properties were implemented (Table 2). We considered the bedrock to have a volumetric mineral content of 97 % and a volumetric water content of 3 %, which implied saturated conditions during the entire simulation. Albedo for limestones was found between approximately 0.22 and 0.32





(Blumthaler and Ambach, 1988) and as light-colored carbonates build up the cliffs, we assumed an albedo of 0.3 in the model setup. An important fitting parameter was the roughness length $z_0$ as performed in Magnin et al., (2017). We set it to a value

of 0.018 m, which represents roughly 1/10 of the height of the surface roughness elements. This fitted well to the rugged rock surface characterized by joint systems (Fig. 2). We set the albedo for the horizontal ground surface to 0.15, for water surfaces to 0.1 and the albedo for ice and snow to a relatively low value of 0.55, as the highest influence of reflected short-wave radiation was expected for spring, when snowmelt decreases the albedo (Table 2).

**Table 2: Model parameters assumed in the simulations.**

| Parameter | | Value | Unit |
|---|---|---|---|
| Albedo rock wall | $\alpha$ | 0.30 | [ - ] |
| Albedo ground | $\alpha_g$ | 0.15 | [ - ] |
| Albedo open water | $\alpha_w$ | 0.1 | [ - ] |
| Albedo melting snow / ice | $\alpha_s$ | 0.55 | [ - ] |
| Emissivity | $\varepsilon$ | 0.97 | [ - ] |
| Roughness length | $z_0$ | 0.018 | [m] |
| Mineral fraction | $mineral$ | 0.97 | [ - ] |
| Water and ice fraction | $waterIce$ | 0.03 | [ - ] |
| Water bucket depth | $d$ | 0.001 | [m] |

Atmospheric forcing was provided by the AROME-Arctic weather model, which is a regional high-resolution, non-hydrostatic numerical weather prediction system for the European Arctic (Müller et al., 2017). It is based on HARMONIE-AROME as part of the ALADIN-HIRLAM system, which provides short-range weather forecasts for Northern and Southern European

countries (Bengtsson et al., 2017; Seity et al., 2011). Archive files of atmospheric data are available since October 2015. In 2017, updates were implemented to improve high-resolution weather forecasts over the Nordic regions (Müller et al., 2017). AROME-Arctic operates on a resolution of ~2.5 km grid spacing at 65 vertical levels. We used time series ranging from October 2015 to August 2020 from AROME-Arctic as forcing data for the model. The nearest 2.5 km grid cell of the AROME-Arctic to the required locations was located northeast of Ny-Ålesund in Kongsfjorden (78.9N, 11.98E, 20 m a.s.l.). The selected

grid cell represents the transition between the fjord and the land and therefore, it provides suitable forcing data for the loggers located directly or within a short distance to the shoreline. The driving variables absolute humidity, wind speed, down-welling short-wave and long-wave radiation, air pressure and rates of snowfall and rainfall for this grid cell have been extracted from the archive. Incident solar radiation at the top of the atmosphere was provided by ERA5 (Hersbach, 2016).





The spatial resolution of air temperature given by AROME-Arctic was not sufficient to capture small-scale variabilities. Therefore, we used records from two climate stations to force the model (Boike et al., 2018, 2019; Maturilli, 2020a, 2020b, 2020c, 2020d). The Baseline surface radiation network (BSRN) station in Ny-Ålesund is located in the village center (78.9250N, 11.9300E) with a distance of about 300 m to the coast (Maturilli et al., 2013). The second station at the Bayelva site is located on top of the Leirhaugen hill, which is in 1.3 km distance to the coast (Boike et al., 2018). Using records of two

different stations allowed us to estimate gradients in air temperatures from the coast to environments further inland. For simplicity, we interpolated linearly between the two stations and estimated the air temperature at the rock walls subject to their distance to the open water body of the fjord. As sea ice coverage enlarges the distance to the open water body, we added an additional mean distance (Table 3), estimated by analysis of the webcamera time series from the mountain Zeppelinfjellet (Pedersen, 2013).


**Table 3: Distances to the open water body used for the linear interpolation of air temperature and logger locations where the distances are applied to.**

| Site | Logger | Distance [m] |
|---|---|---|
| Station Bayelva | - | 1300 m |
| Station Ny-Ålesund | - | 300 m |
| Non-coastal loggers | RW01-RW03 | 600 m |
| Coastal loggers | RW04-RW08 | 0 m |
| Ice cover in the bay | RW08 | +300 m |
| Ice cover in the fjord | RW01-RW08 | +600 m |

We used water temperature of Kongsfjorden recorded by the AWIPEV underwater observatory in 12 m depth to estimate the
long-wave heat emission of the water body. The data provides a time series of water temperatures for the entire period from October 2015 to August 2020 with a resolution of one hour (Fischer et al., 2018a, 2018b, 2018c, 2019).

We simulated long-term climate impacts of three different representative concentration pathways (RCP) RCP2.6, RCP4.5 and RCP8.5 (van Vuuren et al., 2011) for coastal and non-coastal settings. For the period 1980 to 2019, we used forcing data of
the ERA Interim Reanalysis, while the years 2020 to 2100 were created using an anomaly approach based on CMIP5 projections of Community Climate System Model (CCSM4) (NCAR, 2016). Therefore, decadal monthly anomalies were derived from the CCSM4 projections using a reference period from 2009 to 2019 which were then applied to the Reanalysis data of the same period (see Koven et al., 2015).

To account for small-scale variabilities between coastal and non-coastal rock walls, we calculated a linear regression using the
AROME forcing data. Others parameters were simplified due to a lack of information: sea temperature was set to a constant value of 2.53 °C, which is the mean sea temperatures of the analyzed period 2016 to 2020. Sea ice was assumed in the months





February to May until the year 2005 (Gerland and Hall, 2006). Despite these uncertainties, these steps allowed us to analyze possible trajectories for the future developments of the rock wall thermal regime.

**3.5 Model scenarios**

In this study, we considered several scenarios to represent the thermal conditions at the selected rock walls (Table 4). We varied the source of forcing air temperature, the source for heat emission and the albedo of the foot of the slope to account for the different locations. The *non-coastal scenario* was controlled by conditions either with or without snow at the foot of the slope resulting in temporarily changing albedos for snow and terrain. The *frozen bay scenario* showed temporarily frozen seawater leading to changes in the source of heat emission and albedo of the foot of the slope. The *open fjord scenario* was

characterized by a predominantly unfrozen fjord during the entire year and had consequently no varying parameters in most of the simulation time. Short periods of a frozen layer occur temporarily so that the model parameters were modified the same way as in the *frozen bay scenario*. Daily frozen conditions were estimated by analyzing webcamera time series from the mountain Zeppelinfjellet, which provides photos of Ny-Ålesund and the adjacent coastline every ten minutes (Pedersen, 2013).

**Table 4: Model scenarios and corresponding settings with the three varying parameters source for air temperature, source for long-wave heat emission and albedo of the foot of the slope.**

| Scenario name | Representation | Surface state | Albedo | Long-wave radiation computed by |
|---|---|---|---|---|
| Non-coastal scenario | RW01, RW02, RW03 | No snow | Terrain = 0.15 | $T_{air}$ |
|  |  | Snow | Ice / snow = 0.55 |  |
| Open fjord scenario | RW04, RW05, RW06, RW07 | Unfrozen | Water = 0.1 | $T_{sea}$ |
|  |  | Frozen | Ice / snow = 0.55 | $T_{air}$ |
| Frozen bay scenario | RW08 | Unfrozen | Water = 0.1 | $T_{sea}$ |
|  |  | Frozen | Ice / snow = 0.55 | $T_{air}$ |

**4 Results**

**4.1 Measurements of rock surface temperatures**

Mean annual temperatures (Sep–Aug) as well as mean temperatures in the winter season (Dec–Feb) are given in Table 5. For

the measurement period 2016 to 2020, all logger record below-freezing mean annual rock surface temperatures (MARST) with values between -0.6 °C (RW06 in 2017/18) and -4.3 °C (RW02 in 2019/20). The MARST typically vary up to several degrees between the recorded years, with 2017/18 being the warmest year. Measurements in 2019/20 show the lowest MARST, which is related to a comparatively cold winter (Dec–Feb) and spring (Mar–May) season (Table 5).



Minimum and maximum daily RST are found between -24.2 °C (RW03) and 18.9 °C (RW06). The variability of daily RST
show a higher frequency in summer and higher amplitudes in winter. Fluctuations of RST are especially pronounced for non-
coastal rock walls during the cold periods of the year, while the signal at coastal cliffs at the open fjord is dampened in the
same time.

**Table 5: Measured MARST and mean RST in the winter season (Dec–Feb) for all locations RW01 to RW08. Lack of data results**
**from either snow cover on the logger or missing records. MARST are coldest in non-coastal settings (RW01 – RW03). Mean RST in**
**winter season are found to be coldest in non-coastal settings, closely followed by settings in the bay (RW08), while settings at the**
**open fjord show highest RST (RW04 – RW07).**

| Location | Site | Entire year | | | | Winter: Dec-Feb | | | |
|---|---|---|---|---|---|---|---|---|---|
| | | 2016/17 | 2017/18 | 2018/19 | 2019/20 | 2016/17 | 2017/18 | 2018/19 | 2019/20 |
| RW01 | Non-coastal | - | -1.5 | - | - | -8.1 | -6.7 | -9.0 | -13.1 |
| RW02 | Non-coastal | -2.2 | -1.8 | -2.4 | -4.3 | -8.5 | -6.6 | -9.6 | -13.5 |
| RW03 | Non-coastal | -1.8 | -2.0 | -2.4 | -4.1 | -9.2 | -7.2 | -10.0 | -13.8 |
| RW04 | Open fjord | - | -1.0 | -2.1 | -3.6 | - | -5.1 | -8.7 | -12.0 |
| RW05 | Open fjord | -0.9 | -0.8 | - | - | -6.6 | -5.2 | -7.2 | -10.4 |
| RW06 | Open fjord | - | -0.6 | - | - | - | -5.0 | -8.5 | -11.5 |
| RW07 | Open fjord | - | -0.8 | - | -3.6 | - | -4.6 | -7.6 | -11.1 |
| RW08 | Bay | -1.4 | -0.9 | -2.1 | - | -8.6 | -5.8 | -9.7 | -13.1 |

We emphasize that loggers at the coastal cliffs record higher MARST than loggers in non-coastal rock walls with a mean
difference of MARST of 1.0 °C (Table 5), although the loggers are located in just about 1.5 km distance (Fig. 1) at similar
elevations. The setting is especially important in winter and spring season and RST differences account for 1.5 °C to 2.2 °C in
these time periods. The lower the temperatures, the larger is the temperature difference between these two settings, which is
apparent in Fig. 4a.

Besides these observation in RST, time series of station data show that higher air temperatures are recorded for the BSRN
station in Ny-Ålesund compared to the Bayelva site further inland. During the year 2017/18, the air temperature difference
was 0.9 °C with the highest differences of 1.6 °C during winter season and 1.5 °C during spring season.

We highlight that RST values in the Thiisbukta bay are significantly lower in winter than RST at the coastal cliffs even though
they are all located at the shoreline. This is especially true for periods where the bay is characterized by an ice layer on the
water (Fig. 4a), but can also be observed for unfrozen conditions in the bay. If not only the bay is frozen, but widespread sea
ice occurs in the fjord, RST values in all three settings show about the same temperatures (Fig. 4b).



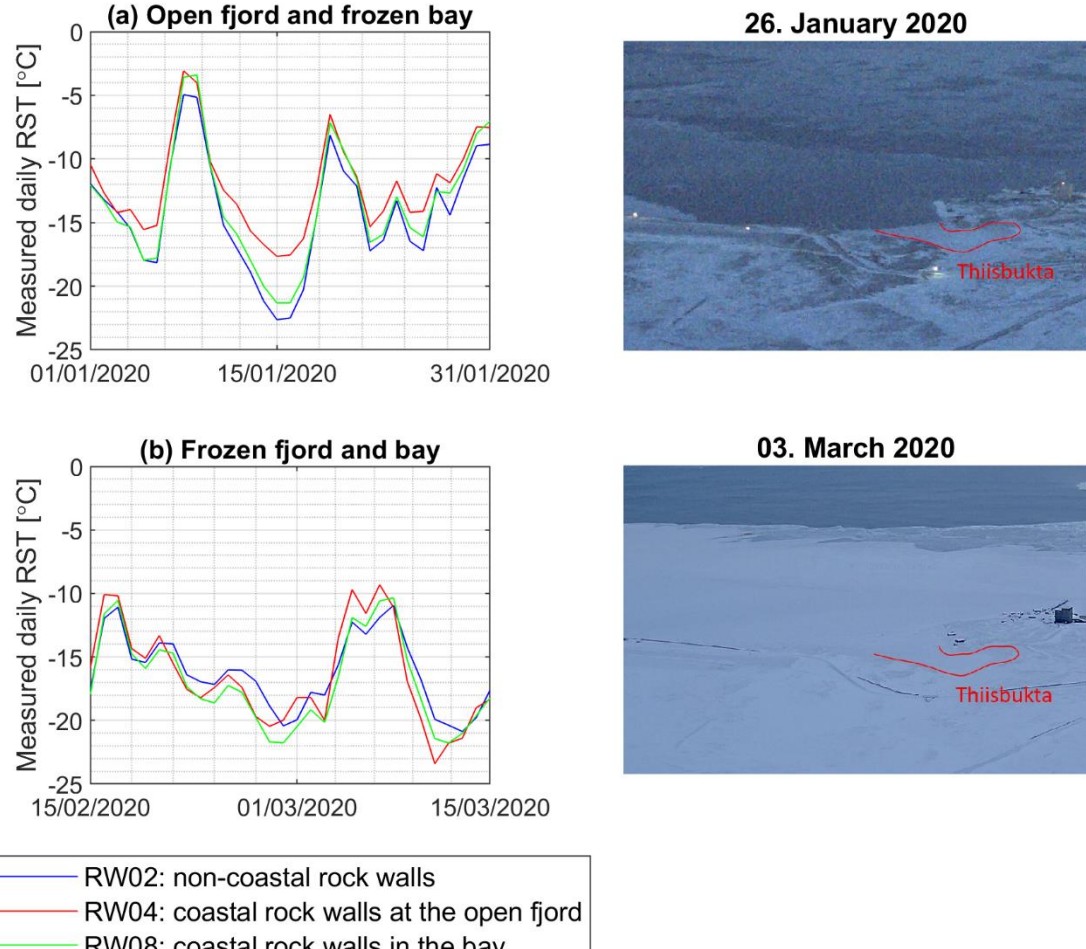

**Figure 4: Mean measured daily RST for two different conditions in Kongsfjorden: (a) The time period 01.01.-31.01.2020 was mainly characterized by a frozen bay and an open fjord. RST at coastal cliffs show higher values than RST at the non-coastal rock walls**

**and at rock walls in the bay. The most pronounced differences are found with RST below -10 °C. (b) At the time period 15.02.-15.03.2020, the bay and the fjord were predominantly frozen. In this case, no clear differences could be observed between the three settings. RW02, RW04 and RW08 were selected as they have the same aspect but different settings.**

## 4.2 Model validation

We compared monthly average values of measured rock surface temperature RST to the model results of the *non-coastal scenario* (RW01, RW02, RW03), the *open fjord scenario* (RW04, RW05, RW06, RW07) and the *frozen bay scenario* (RW08). The measured RST was reproduced closely with the applied model setup, especially for temperatures near freezing point (Fig. 5). Besides the visually good agreement of Fig. 5, a root-mean-square error (RMSE) below 1.2 °C, the bias (-0.5 °C to





0.4 °C), the coefficient of determination $R^2$ (above 0.97) and the Nash-Sutcliffe efficiency NSE (above 0.96) for all locations
confirmed a good reproduction of the measured data (Table 6).

**Table 6: Summary statistics of the model validation including RMSE, bias, $R^2$ and NSE. The statistics were calculated for all locations RW01 to RW08, comprising each entirely recorded month for the measurement periods stated in Table 1.**

| Location | Site | RMSE | bias | $R^2$ | NSE |
|---|---|---|---|---|---|
| RW01 | Non-coastal | 0.8 | 0.0 | 0.989 | 0.986 |
| RW02 | Non-coastal | 1.0 | 0.3 | 0.984 | 0.981 |
| RW03 | Non-coastal | 0.9 | 0.4 | 0.989 | 0.986 |
| RW04 | Open fjord | 0.7 | 0.3 | 0.991 | 0.988 |
| RW05 | Open fjord | 1.1 | -0.5 | 0.978 | 0.968 |
| RW06 | Open fjord | 1.0 | -0.1 | 0.980 | 0.980 |
| RW07 | Open fjord | 1.1 | 0.2 | 0.980 | 0.969 |
| RW08 | Bay | 1.2 | 0.3 | 0.992 | 0.977 |

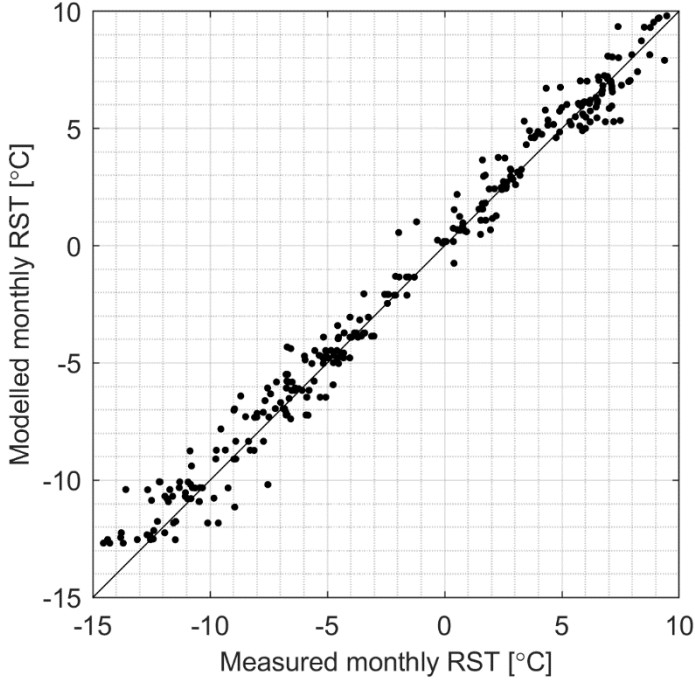


**Figure 5: Mean modelled vs. measured monthly RST for all locations RW01 to RW08, comprising each entirely recorded month for the measurement periods stated in Table 1.**



### 4.3 The influence of open water and sea ice on RST

The model results are in good agreement with the in situ measurements and corroborate the pattern of open water and sea ice
influence on RST.

Below-freezing MARST are modelled for all locations, ranging from -0.3 °C (RW06 in 2017/18) to -3.9 °C (RW02 in 2019/20)
with the warmest year being 2017/18 and the coldest year being 2019/20. The lowest MARST are modelled at rock walls in
the *non-coastal scenario*, while the *open fjord scenario* produces the highest MARST.

The model results show differences in MARST according to the exposition of the rock wall: In the *open fjord scenario*, the
lowest MARST in 2017/18 is found on the north-facing rock wall RW05 (-0.9 °C), while the highest MARST is calculated for
RW06 facing east-north-east (-0.3 °C). Accordingly, the model results with a similar model setup are dependent on the
orientation of the rock wall.

In the model results, we find that temporarily occurring ice cover on the fjord results in lower RST at the nearby rock walls.
For time periods with a frozen bay, but no sea ice in the open fjord, only RW08 is affected. Model results show ca. 1 – 1.5 °C
colder RST compared to the other rock walls at the shoreline but they are still warmer than the modelled RST in non-coastal
settings. However, the results indicate that days with a widespread sea ice extent in the fjord lead to similar RST in all locations
(Table 7).

**Table 7: Modelled mean RST with frozen conditions in the bay or sea ice in the fjord. Frozen conditions in the bay lead to a local cooling of RW08, while sea ice result in similar RST for all settings. RW02, RW04 and RW08 were selected as they have the same aspect but different settings.**

| Location | Site | Dec | Jan | Feb | Mar | Apr |
|---|---|---|---|---|---|---|
| | | Mean RST of the days with a frozen bay | | | | |
| RW02 | Non-coastal | -11.8 | -13.5 | -13.9 | -15.0 | -7.6 |
| RW04 | Open fjord | -10.2 | -10.6 | -11.7 | -13.4 | -6.8 |
| RW08 | Bay | -11.2 | -12.3 | -13.0 | -14.4 | -7.2 |
| | | Mean RST of the days with widespread sea ice in the fjord | | | | |
| RW02 | Non-coastal | - | - | -17.3 | -15.0 | -10.9 |
| RW04 | Open fjord | - | - | -17.2 | -15.0 | -10.9 |
| RW08 | Bay | - | - | -17.2 | -15.0 | -10.9 |

Lower RST values under frozen conditions can be traced back to three different factors: While (1) lower air temperature and
(2) the lack of heat emission from the ocean lead to a cooling of RST, (3) the reflection of short-wave radiation on the ice layer
increases RST as an additional energy source. The amount to which these factors influence the decrease in RST between the
*open fjord scenario* and the *frozen bay scenario* is given in Fig. 6. Between December and February, air temperature and the





lack of radiative heating are the dominant factors, while reflected short-wave radiation plays no role. In March, the influence of reflected short-wave radiation increases as polar night conditions end.

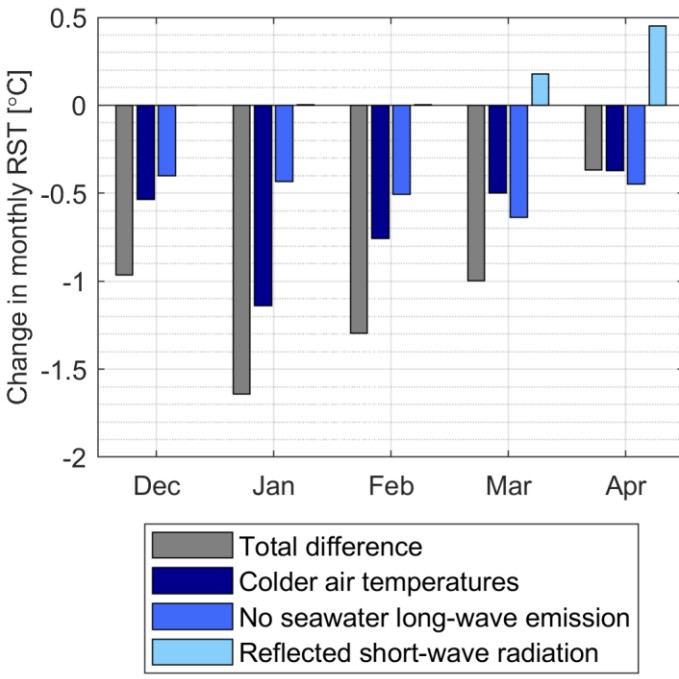


**Figure 6: Temperature difference between the *open fjord scenario* (reference) and different model scenarios. "Colder air temperature": as reference, but using colder air temperature as distance to the open water body is enlarged; "No seawater long-wave emission": as reference but using air temperature instead of seawater temperature; "Reflected short-wave radiation": as reference but assuming ice albedo for the surrounding terrain; "Total difference": *frozen bay scenario* combining all three effects.**


### 4.4 The surface energy balance

Individual fluxes of the surface energy budget in the different seasons are given in Fig. 7 (Winter = Dec–Feb; Spring = Mar– May; Summer = Jun–Aug; Fall = Sep–Nov) with positive fluxes directed towards the surface. For comparison of the different scenarios, the fluxes are calculated for vertical rock walls with an aspect of 40° (~NE), which comprises model runs of RW02,

RW04 and RW08.





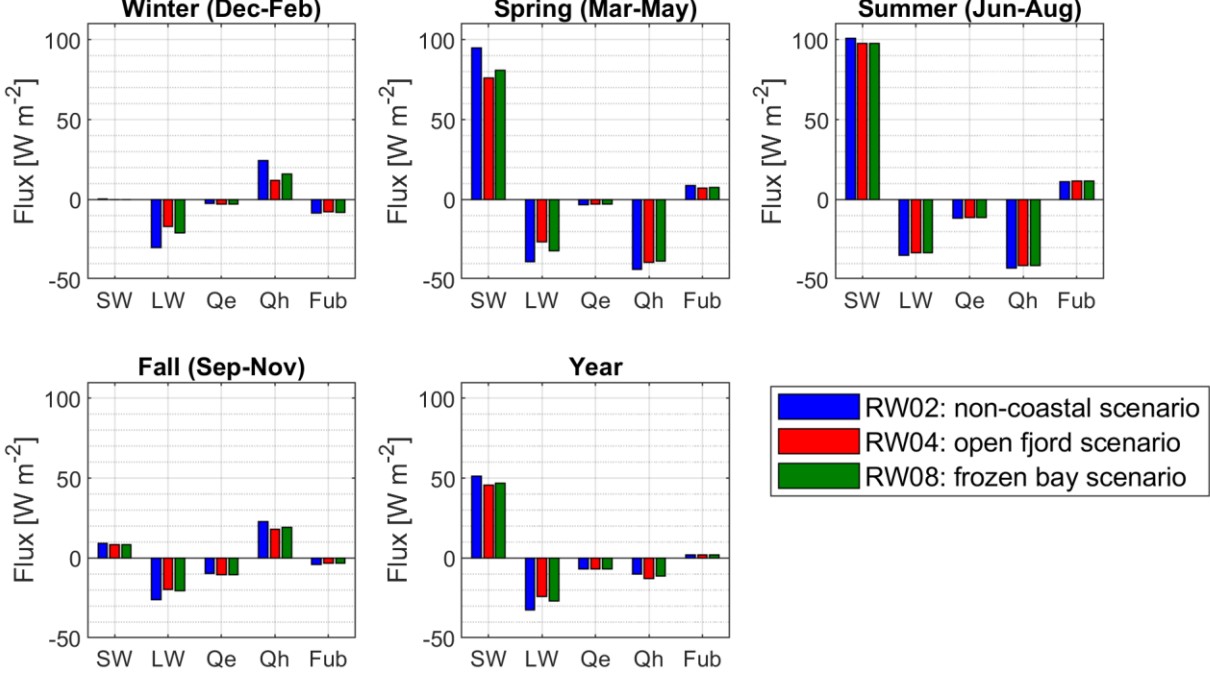

**Figure 7: Surface energy balance for the seasons of the year for a vertical rock wall with an aspect of 40° (RW02, RW04 and RW08). Most pronounced differences in the scenarios are found in winter and spring, while summer and fall show similar fluxes. SW = net short-wave radiation; LW = net long-wave radiation; Qe = latent heat flux; Qh = sensible heat flux; G = ground heat flux.**


During winter which mostly coincides with polar night conditions (25 October to 14 February), short-wave radiation is zero or only reaches very small values. During this period, the system loses energy mainly due to negative net long-wave radiation, which is especially pronounced for the *non-coastal scenario,* followed by the *frozen bay scenario*. The loss of energy is opposed by positive sensible heat fluxes, representing a warming of the surface and a cooling of the atmosphere. Strong

sensible heat fluxes are associated with high wind speeds and high temperature differences of air and rock wall. Compared to the other terms of the SEB, the negative ground heat flux leading to ground cooling is only small.

In spring, net short-wave radiation increases significantly and becomes the dominant energy source with highest energy input for the *non-coastal scenario* and the lowest for the *open fjord scenario*. Long-wave radiation counteracts this process and cool

the surface with highest fluxes in the *non-coastal scenario*. Besides, sensible heat fluxes contribute to the energy loss with slight differences in the scenarios. However, sensible heat fluxes and emitted long-wave radiation cannot compensate the incoming energy by short-wave radiation and RST as well as ground heat fluxes start to increase, especially as no energy is used for melting due to snow-free conditions.





The summer period is characterized by similar fluxes of the surface energy balance in all scenarios. The warming of RST continues due to strong short-wave radiation as the main energy source. Energy is lost by long-wave radiation and sensible heat fluxes but also latent heat fluxes cool the surface. However, these fluxes cannot compensate the energy input. Consequently, the ground heat flux increases even more, leading to a seasonal thawing of the active layer.

During fall, net short-wave radiation decreases rapidly due to shorter days, but sensible heat fluxes turn positive, acting as an energy source again. The loss of energy by long-wave radiation is slightly higher for *non-coastal scenarios* and ground heat fluxes are close to zero, indicating the turn to refreezing of the active layer.

In the course of a year, short-wave radiation is naturally the main source of energy to the system, while most energy is lost by
long-wave radiation. Sensible heat fluxes warm the surface in fall and winter, while they cool in spring and summer. Latent heat fluxes are of minor importance, reflecting the small water holding capacity of the rock surface assumed in the model. Net ground heat fluxes are close to zero.

## 4.5 Simulations of future climate change scenarios

The past and future simulations of different RCPs show an increase in MARST for both the *non-coastal* and the *open fjord*
*scenario* (Fig. 8). Between 1980 and 2020, MARST increases by several degrees and the MARST difference of the *non-coastal scenario* and the *open fjord scenario* is significant. We emphasize that in this period winter sea ice loss has been drastic in Kongsfjorden, going from a normally frozen fjord to a normally open fjord. Thus, in reality, the actual warming may have even been higher than either of the scenarios suggests. Figure 8 clarifies, that the current measurement period represents relatively warm years in the occurring fluctuations of MARST and a drop to colder MARST in 2020.
During the years 2020 to 2080, all three pathways show slightly increasing MARST. The influence of the different rock wall locations decreases with time and consequently, MARST values of the *non-coastal* and *open fjord scenario* become more aligned. After 2080, RCP2.6 and RCP4.5 are stabilized with MARST between -4 °C and 0 °C, while RCP8.5 shows further increasing MARST reaching mean annual values up to 2 °C.

The model results suggest that a large part of the warming in RST has already happened until the year 2020, while the
prospective increase of RST in the 21st century will get the permafrost close to thawing.





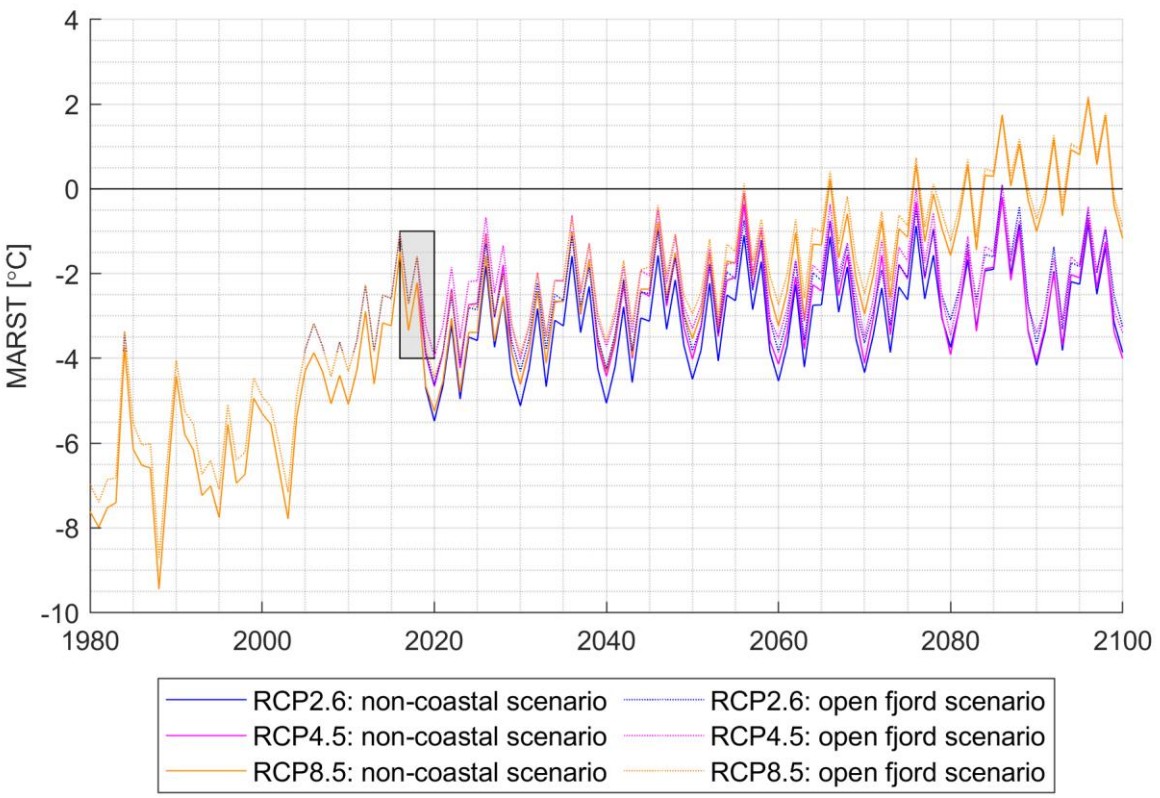

**Figure 8: MARST of past and future simulations with the settings of the *non-coastal scenario* and the *open fjord scenario* with RCP2.6, RCP4.5 and RCP8.5. The grey box shows the measurement period.**

Furthermore, simulations of the RCPs suggest an effect on the active layer thickness ALT (Fig. 9). Between 1980 and 2010, ALT is about 2 m, with a slightly deeper active layer for the *open fjord scenario*. After 2010, the ground begins to thaw deeper during summer season, but the future evolution varies between the three simulated RCPs: RCP2.6 is characterized by a slight increase in ALT and a stabilization between 2.5 m and 3.5 m after 2080. RCP4.5 shows a similar trend with ALTs between 3.0 m and 4.5 m at the end of the century. However, the simulation of RCP8.5 results in a significant increase of ALT below

8 m and with no apparent stabilization effect. Moreover, a talik is developed after 2095, implying that the cold winter seasons do not lead to a freezing of the entire ground column anymore.





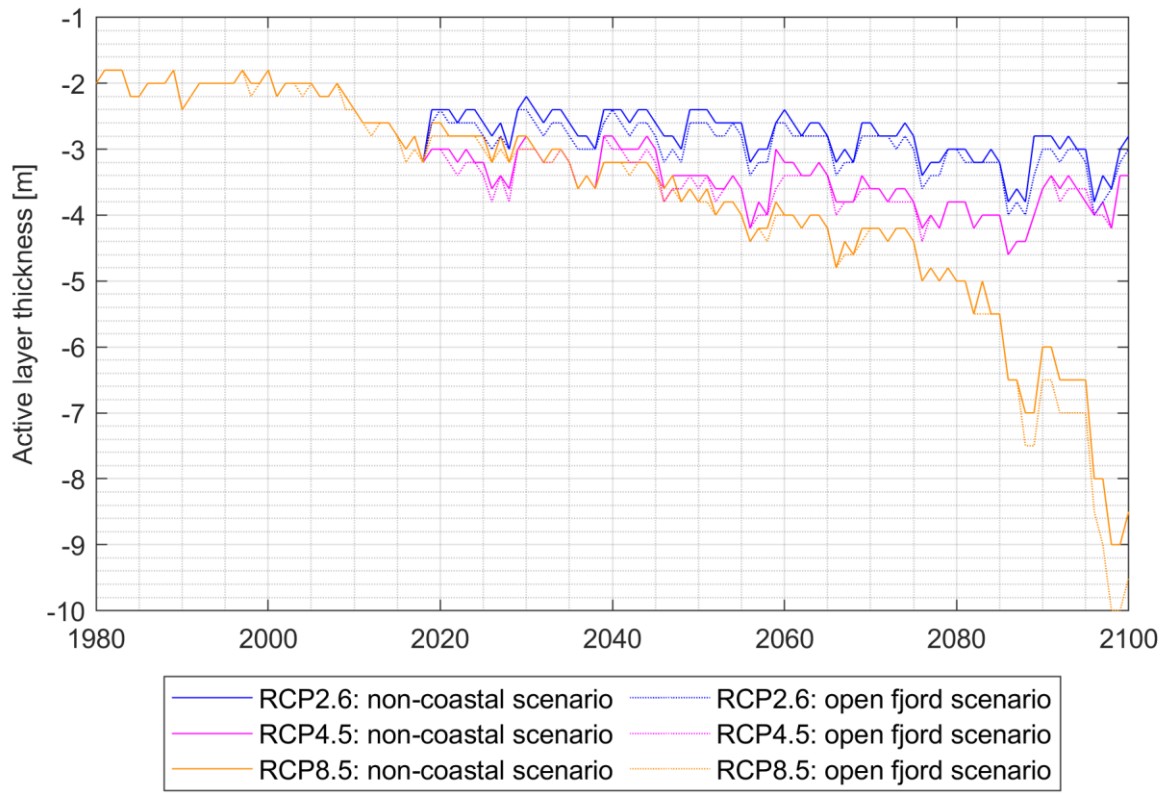

**Figure 9: Active layer thickness of past and future simulations with the settings of the *non-coastal scenario* and the *open fjord scenario* with RCP2.6, RCP4.5 and RCP8.5.**

## 5 Discussion

### 5.1 Model uncertainties

Our model setup contains uncertainties regarding unknown model parameters, which were estimated using literature and calibrating the model. Especially rock (surface) parameters could be improved by a more precise analysis of the lithological characteristics.

A critical point is the assumption of a neutral atmospheric stratification perpendicular to the vertical rock wall which must be regarded as a first order approximation. This leads to uncertainties in near surface turbulent exchange of the vertical wall as micro-topography and changing weather conditions can influence the movement of the air parcels. Wind profiles perpendicular to the wall as well as measurements of air temperature and humidity could help to estimate the importance of this error source.





However, we scaled the roughness length $z_0$ to compensate these assumption and to fit the modelled surface temperatures to the observed values.

Another error source is the estimation of air temperature at the logger location. The linear interpolation between the two stations and the dependency on the distance to the open water body of the fjord is a coarse approximation at best which does not take

the wind direction, the atmospheric boundary layer structure and local micro-climate into account. Air temperature measurements at the shoreline could help to improve the quality of the forcing data.

### 5.2 The influence of open water and sea ice on RST

In the time period 2016 to 2020, all records display negative MARST which indicates permafrost conditions at all locations (Table 5). Svalbard lies in the continuous permafrost zone (Brown et al., 1997; Obu et al., 2019) and deep permafrost is

observed in the area, e.g. within the abandoned mine shafts (Liestøl, 1977) and in boreholes (Christiansen et al., 2010).

Two main factors control the RST difference during winter season between the relatively warm *open fjord scenario* and the relatively cold *non-coastal scenario*: (i) Air temperature gradients from the coast to the inland play an important role. They have a pronounced effect on the surface energy balance, especially on turbulent fluxes, and result in higher RST at the coastal

settings. Furthermore, (ii) incoming long-wave radiation leads to higher RST values at the coastal cliffs. The rock walls receive energy by long-wave radiation emitted by the surfaces in the field of view, which is controlled by seawater temperature for open fjord settings and by air temperature for non-coastal rock walls. As the seawater has significantly higher temperatures than the air during winter, the energy input is larger at the coastal cliffs.

Thick snow deposits (> 1 m depth) can effectively insulate the ground resulting in higher RST during winter (Haberkorn et al., 2015). This is also true for Ny-Ålesund where ground surface measurements and model results indicate higher mean annual ground surface temperatures (MAGST) for planes with thick snow covers as documented by Gisnås et al., (2014). However, thin snow cover (< 0.5 m depth) can lead to a lowering of RST as low air temperatures can still affect the rock while the high albedo of the snow reflects large parts of solar radiation (Haberkorn et al., 2015; Magnin et al., 2017). Both the warming and

the cooling effects are apparent in RST measurements at those logger positions, which are temporarily covered by snow (Fig. 3). Furthermore, a direct comparison of MARST in the largely snow-free cliffs with MAGST measured in near-horizontal tundra settings close to the Bayelva station (using the measurement setup of Gisnås et al., (2014)) suggest that MARST at coastal cliffs can be even warmer than MAGST under a thick snow cover.

While an ice cover in the bay leads to locally decreased RST in the bay, a widespread sea ice extent results in lower RST for all settings. Maximum sea ice coverage is significantly reduced since 2006 and a shorter sea ice season is observed since 2002 (Johansson et al., 2020). Hence, it can be assumed that RST at the coastal cliff has increased since then in winter and spring





season. Three main factors could be identified which influence the model results: During polar night conditions, (i) air temperature gradients between the open water body and inland and (ii) radiative heating by comparatively warm seawater

strongly affect RST, leading to lower RST for frozen conditions. When incoming solar radiation increases again in March, (iii) the increased reflection of short-wave radiation on the ice cover can counteract these processes to a certain extent.

### 5.3 The surface energy balance

The components of the surface energy balance are estimated for different seasons of the year. In summer and fall, fluxes are largely similar for the different scenarios, while significant differences can be noticed in winter and spring (Fig. 7).


Net short-wave radiation is the dominant source of energy for all scenarios in spring and summer due to midnight sun conditions. The flux is especially strong, when solar radiation is strongly reflected by surrounding sea ice or snow cover, in addition to the direct and diffuse short-wave radiation. This can be seen in spring in the *non-coastal scenario* (reflection on snow) and the *frozen bay scenario* (reflection on ice). As the bay is not continuously frozen, the effect is less pronounced in

the *frozen bay scenario*. In the *open fjord scenario*, reflection of short-wave radiation plays a minor role as a result of the low albedo of seawater.

The system loses energy by net long-wave radiation during the entire year and the differences between the scenarios are most pronounced in winter and spring. The small net long-wave radiation in the *open fjord scenario* can be explained by higher

incoming long-wave radiation through emission of the relatively warm seawater. During summer, the temperature difference between seawater and air is smaller which limits the influence compared to the cold seasons. However, sensible heat fluxes are a major component and play an important role during the entire year. In winter and fall, they warm the surface and cool the atmosphere, while this process is reversed in spring and summer. In winter, sensible heat fluxes are larger in the *non-coastal* and *frozen bay scenario* compared to the *open fjord scenario*. As roughness lengths and wind speed are assumed to be

the same, this effect can be traced back to larger temperature differences of air and rock wall. Therefore, the air temperature gradients in the surroundings of Ny-Ålesund intensify the sensible heat transfer to the surface and influence the surface energy budget of the rock walls significantly. Latent heat fluxes play a minor role in the energy budget as just a small amount of water can be stored in the vertical bedrock that is available for evaporation.

### 5.4 Future climate change scenarios

At the costal cliff sites, our simulations suggest that MARST have significantly increased in the last 40 years. The increase of MARST is especially pronounced since the year 2000. This is in line with increasing ground temperatures, observed from 1998 to 2017 at the Bayelva station close to the setting described in this study (Boike et al., 2018). A further increase of MARST is predicted for all pathways with positive values for RCP8.5 at the end of the century. The main reasons are increasing air temperatures and long-wave radiation in the course of the 21[st] century.





Another important effect is the reduced difference of the *non-coastal scenario* and the *open fjord scenario* with ongoing permafrost warming. This can be explained by a convergence of seawater temperature and air temperature during the winter season in the assumed model forcing, which likely reflects true conditions in a future ice-free Arctic. As a consequence, the influence of the relatively warm seawater on the radiation budget and coastal air temperatures becomes less important under a warmer climate and RST of non-coastal and open fjord settings become more similar.


Moreover, the impact of a changing climate becomes visible in deeper layers of the ground, e.g. through a deepening of the active layer. However, these model results must be interpreted carefully as additional warming from the top of the cliff must be considered, depending on the geometry of the cliffs. Therefore, the presented model results for deeper layers are likely biased for the investigated small rock walls, while they might be applicable to higher cliffs in the close surroundings of Ny-475 Ålesund.

### 5.5 Coastal cliffs in the high Arctic – a future geohazard?

Our model results indicate a significant warming of permafrost temperatures and a deepening of ALT in the 21$^{st}$ century, a trend that can affect the mechanical erosion of the rock walls. The time and depth spend in the optimal thermal range for frost cracking between -3 °C and -8 °C may change (Anderson, 1998; Hales and Roering, 2007) and longer durations of open-water 480 season and a correlated interaction of water with the shoreline can enhance coastal erosion (Barnhart et al., 2014). Besides, warming of permafrost temperatures can lead to further destabilization as rock- and ice-mechanical properties are modified (Krautblatter et al., 2013). At the seashore, loss of sea ice may intensify the expected degradation of permafrost and result in enhanced slope instabilities.

In this study, the thermal regime of relatively low coastal cliffs is investigated. Indeed, similar processes can also affect much higher coastal cliffs on Svalbard, for example in the Mesozoic rocks in the Isfjorden area (Etzelmüller et al., 2003). As they are mostly not entirely vertical, corresponding sky view factors might be larger, resulting in a slightly smaller influence of long-wave radiation emitted by seawater and reflected short-wave radiation from sea ice. However, warmer coastal air temperatures will have an impact independent on inclination of the slope. As a consequence, shorter durations of sea ice cover 490 will likely lead to an enhanced warming of permafrost in high rock cliffs in Svalbard.

Failures of coastal rock slopes can impact the water body and trigger displacement waves along shorelines (Hermanns et al., 2006). Prominent examples are the landslide in Paatuut / West Greenland in 2000, creating a tsunami with a run-up height of up to 50 m (Dahl-Jensen et al., 2004) and the rock avalanche in Tåfjord / Norway in 1934, producing an up to 62 m high 495 displacement wave (Hermanns et al., 2006). Despite different geological and climatological settings, an occurrence of rock slope failures causing tsunamis cannot be excluded for Svalbard. Large active rock slope instabilities over water bodies have been detected, like the Forkastningsfjellet rockslide in Isfjorden of circa 300.000 m$^3$, which is located in just 8 km distance of



Longyearbyen with 2200 inhabitants (Kuhn et al., 2019). Due to permafrost degradation, rock slope failures in the high Arctic might become more likely in future.

## 6 Conclusion

In this study, we present measurements of rock surface temperatures (RST) from steep coastal and non-coastal cliffs in the high Arctic setting of Ny-Ålesund, Svalbard, comprising data from 2016 to 2020. The permafrost model CryoGrid 3 is applied for this thermal regime with an adapted parametrization of radiative forcing: The slope angles and aspects of the rock walls have been taken into account as well as additional heat sources like long-wave emission from seawater and reflecting short-wave radiation on snow and ice covers. With our measurements and model results, we can draw the following conclusions:

- Measured RSTs in coastal cliffs are up to 1.5 °C higher during the winter season than the non-coastal rock walls. Model results suggest that this results from slightly higher air temperatures at the coast compared to inland locations as well as from the continuous energy input by long-wave radiation from the relatively warm seawater.
- When the sea adjacent to the coastal cliff is covered by sea ice, coastal RSTs are decreased and closely match RST at inland locations. This can be explained by lower air temperatures and disabled long-wave emission of seawater. Reflection of short-wave radiation on the ice cover counteracts this process, but is only effective when polar night conditions end. As a consequence, sea ice loss in Kongsfjorden is expected to increase RST on coastal cliffs during winter.
- Simulations for future climate conditions show an increase of mean annual rock surface temperatures MARST with a stabilization between -4 °C and 0 °C for RCP2.6 and RCP4.5 and a further warming to above-freezing MARST for RCP8.5 at the end of the century. Furthermore, the model predicts a deepening of the active layer for all RCPs.

## Appendix

### Appendix A: Projection of direct short-wave radiation on inclined planes

We determined the direct short-wave radiation perpendicular to an inclined slope $S_{dir\_slope}$ by projecting the direct short-wave radiation on the horizontal $S_{dir}$ as following:

$$S_{dir\_slope} = \frac{S_{dir}}{\cos(\delta_{sun\_hor} * \frac{pi}{180})} * \cos(\delta_{sun\_slope} * \frac{pi}{180}). \tag{A1}$$

with $\delta_{sun\_hor}$ being the angle between the solar rays and the normal on the horizontal plane and $\delta_{sun\_slope}$ being the angle between the solar rays and the normal on the inclined slope. The valid solar zenith angle was set to 85° excluding the very

early sunrise and sunset, as this would lead to highly overestimated energy input on a vertical wall due to nearly vertical solar rays.

**Appendix B: The water bucket approach**

We based the infiltration of water on a water bucket approach. As long as the uppermost grid cell is unfrozen, water could infiltrate and the water content of the grid cell increased. After reaching saturation, excess water was removed by surface
runoff. We calculated the potential evaporation with the turbulent latent heat flux based on the Monin-Obukhov similarity theory (Monin and Obukhov, 1954). We weighted the amount of evaporated water with the water content of the grid cell, taking into account that water can be evaporated more easily under saturated conditions.

**Code availability**

The source code is available at: http://doi.org/10.5281/zenodo.4277514

**Data availability**

Data from field observations are currently in the process of publication in NIRD Research Data Archive. In the meantime, the data are available from the corresponding authors.

**Author contribution**

JS designed the concept of the study, conducted fieldwork, developed the model code and prepared the manuscript including
all tables and figures. SW provided help, ideas as well as organizational and technical support at all phases of the study. BE contributed with subject-specific background and advice during the preparations. SW and FM designed the observation array and conducted fieldwork. TVS provided code for retrieving the AROME forcing data. ML helped with the code development and provided forcing data for future simulations. JB provided climate data of the Bayelva station. All authors contributed to the final manuscript with input and suggestions.

**Competing interests**

The authors declare that they have no conflict of interest.



**Acknowledgement**

We acknowledge funding by Nunataryuk (EU grant agreement, no. 773421), FrostCliff (Research Council of Norway, no. 317378) and the Department of Geosciences, University of Oslo.

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
