# Peer review of "Surface temperatures and their influence on the permafrost thermal regime in high Arctic rock walls on Svalbard"

_The Cryosphere, 2020_

## Referee Comment (RC1) · Anonymous Referee #1 · 5 Jan 2021

General comments:

This paper presents a four-year measurement series of near rock surface temperatures at coastal cliffs or canyon walls close to the Kongsfjord in the Ny Ålesund region, Svalbard and uses the well-known model CryoGrid 3 to calculate the energy balance at the observation sites and discusses the results. The authors show that the coastal sites are in general warmer than the sites a little bit further away from the fjord. This paper is carefully elaborated and shows interesting data and model results, which is an important contribution to a better understanding of the surface energy balance and related heat fluxes at a place where until now not many studies have been carried out

in steep cliffs closely situated to a fjord. In general, before publishing this valuable article some major changes should be considered, which are outlined below:

Line 167, Neglecting influence of snow:

It is somehow understandable that the influence of snow is neglected for the investigated sites as the cliffs are steep and the influence of snow seems according to the authors until now not to be very important. However, there are two important points, which the authors do not consider enough in their paper, although they discuss the influence of snow in the paragraph of the discussing section on line 421 but this discussion cannot be used to justify their main approach of neglecting snow completely in their model approach.

- Firstly, recent studies show clearly that even in steep rock walls, small or even larger amounts of snow can accumulate on rock ledges and influence rock wall temperatures and change heat fluxes considerable (e.g.: Haberkorn et al. 2015, 2016 and 2017) and their logger RW01 and their comment in the caption of table 5 shows that snow seems already today have some influence at certain cliff sites!

- Secondly, performing climate scenarios coupled to model runs of CryoGrid 3 not taking into account snow for future conditions is not at all reasonable as in the future probably snow cover is may going to increase considerable during winter time particularly at sites which are close to a large moisture source (North Atlantic Ocean).

Therefore, I suggest to a) include in CryoGrid 3 a scenario including snow, which should not be a big problem as CryoGrid 3 is already well prepared and b) omit the future scenario runs in this paper (mainly chapter 4.5 and 5.4), as this part does not add any important additional information or only results, which are extremely speculative as the authors admit by themselves on line 474. Particularly as the authors cannot really convincingly show and explain that snow is not going to play an important role in the future. However, a future snow cover at the small cliff sites can strongly change the whole thermal regime (e.g. the canyon site could be filled by more snow and
corresponding snow drift by wind in the future).

Line 121, assumption neutral atmospheric stratification:

It is understandable that the authors try to simplify the very complex processes of the turbulent fluxes. However, assuming that the conditions at the cliff walls could be approached by using for all cases neutral atmospheric stratification is not justified. This assumption would mean that an air parcel moving close to the cliff would always have the same temperature (and density) as the surroundings at this position. This seems to be justified only for certain conditions during the year. The authors also justify in the discussion section their approach by tuning the roughness length until the model fits the observed values. This is reasonable to do. However, their estimated roughness length is, at least after my knowledge of such values and having seen the pictures of the measurement sites in the paper, at least one order of magnitude smaller what should be probably taken as roughness length for these specific observation sites.

Line 107, table 1 setting of surface temperature loggers:

The temperature loggers are located all at expositions of NE (except two of them ENE and N). It would be interesting to see the influence of the different expositions, which could be easily modelled by CryoGrid 3. I would assume that at this latitude the expositions do not play a very important role but it would be an interesting questions which could be answered by CryoGrid 3.

Line 477, coastal cliffs in the high Arctic – a future geohazard:

This chapter does not add any important new information to the main topic of the paper. Please delete this section.

Specific comments:

1. Line 31: better use: warming of atmosphere than warming of climate. The atmosphere can warm but the climate can only change but not warm.

[Figure]

2. Line 46: may add the new literature from Etzelmüller et al. 2020; Etzelmuller, B., Guglielmin, M., Hauck, C., Hilbich, C., Hoelzle, M., Isaksen, K., Noetzli, J., Oliva, M., Ramos, M. (2020) Twenty years of European Mountain Permafrost Dynamics – the PACE Legacy. Environmental Research Letters 15, 14.

3. Line 50: May add some more literature here such as: Gisnås, K., Westermann, S., Schuler, T.V., Melvold, K., Etzelmüller, B. (2016) Small-scale variation of snow in a regional permafrost model. The Cryosphere 10, 1201-1215. Gisnås, K., Westermann, S., Schuler, T.V., Litherland, T., Isaksen, K., Boike, J., Etzelmüller, B. (2014) A statistical approach to represent small-scale variability of permafrost temperatures due to snow cover. The Cryosphere 8, 2063-2074. Haberkorn, A., Wever, N., Hoelzle, M., Phillips, M., Kenner, R., Bavay, M., Lehning, M. (2017) Distributed snow and rock temperature modelling in steep rock walls using Alpine3D. The Cryosphere 11, 585-607.

4. Line 51: Exposition is not only important for steep rock walls. It is in general important also for less inclined slopes particularly at lower latitudes; already much older literature has shown this.

5. Line 82: please give a value for the altered net short-wave radiation through the decrease in reflection so that a comparison to the value given for the change in downward long-wave radiation can be done.

6. Line 88: Is there no mean annual precipitation available in Ny-Ålesund after 2000?

7. Line 101: Please give some information about temperature logger calibration.

8. Line 104: the expression 'non-coastal rock walls' seems not very adequate chosen as this canyon cliffs are only about 600 m from the coast. In my view a 'non-coastal rock wall' would be several kilometers away from the fjord. Please change the wording.

9. Line 114/115: how is the latent heat effect considered. Please explain or give at least a reference where the reader could get more information.

10. Line 176/177: please give a source for this volumetric ice and mineral content

percentages.

11. Line 185, table 2: please give information about the source of your values you show in table 2. You could add a column in the table and show the references.

12. Line 194/195: Is the effect (transition between fjord and land) you describe here really resolved? I can hardly believe this!

13. Line 203: why do you use the radiation data from AROME-Arctic dataset when you have much better data from the BSRN stations. Please clarify?

14. Line 240, table 4: RW01 is modeled according to table 4, but it is shown that this site is covered by snow in figure 3. Therefore, it is mandatory to include snow in the model scenarios for this logger otherwise you contradict yourself in the paper (see also the general comments about snow).

15. Line 255, table 5: your model does not include snow but you wrote that only one logger is snow covered RW01 (figure 3). Please clarify this as it is very important for your assumption that there is no snow cover at the sites.

16. Line 275, figure 4: why is the variability (daily values?) not higher in comparison to figure 3 where there is much more variability in the same data. please clarify.

17. Line 335, figure 7: What is Fub in the figures? Is this not G as noted in the figure caption?

Literature:

Haberkorn, A., Wever, N., Hoelzle, M., Phillips, M., Kenner, R., Bavay, M., Lehning, M. (2017) Distributed snow and rock temperature modelling in steep rock walls using Alpine3D. The Cryosphere 11, 585-607. Haberkorn, A., Phillips, M., Kenner, R., Rhyner, H., Bavay, M., Galos, S.P., Hoelzle, M. (2016) Thermal regime of rock and its relation to snow cover in steep alpine rock walls: gemsstock, central swiss alps. Geografiska Annaler: Series A, Physical Geography 97, 579-597. Haberkorn, A., Hoelzle,

M., Phillips, M., Kenner, R. (2015) Snow as a driving factor of rock surface temperatures in steep rough rock walls. Cold Regions Science and Technology 118, 64-75.

---

## Referee Comment (RC2) · Alessandro Cicoira (Referee) · 15 Jan 2021

General comments:

This manuscript presents a four-year time series of eight temperature loggers at rock cliffs in the surroundings of Ny- Ålesund. The authors use the model CryoGrid 3 in order to discuss the measurements and resolve the influence of the different components on the energy balance on the observations. In addition, the model is combined with three different representative concentration pathways in order to predict the evolution of rock cliffs temperature and active layer thickness throughout the next century. The measurements advance our knowledge of the energy balance at rock cliffs in the

Arctic, and the model is useful for their discussion. However, I have some concerns regarding the manuscript. In my opinion, the temperature measurements are not up to the state of the art, and the modelling work is promising but could be largely integrated with more simulations: adding the two parts is still not sufficient for a publication.

Regarding the measurements: I could not find any information about the calibration of the temperature loggers. This is a major point of concern and strongly weakens all the sequent results and discussion. Additionally, I don't understand why the measurements have been performed with an accuracy of only 0.5°C. In general, I would like to have an explanation of the sampling strategy, which is to my knowledge not up to the state of the art in this field.

Regarding the modelling: a sensitivity study to the many model parameters would be beneficial to the conclusions of the paper and could, with a proper set up, provide interesting insights in the investigated processes. In the modelling in general, and in particular for the future climate scenarios, the quantification of the uncertainty (related to the climate scenarios and their propagation in the modelling) is required.

Due to the limits of the temperature time series and the current state of the modelling, I suggest to restructure the manuscript in order to provide a more thorough study. Personally, I suggest to focus the manuscript on the modelling part: use the observations for model calibration and then use this to perform a more complete series of synthetic experiments to investigate the energy balance in different conditions. Therefore, I consider the manuscript promising and potentially suited for publication, but I suggest some major revisions prior to publication. A short list of specific comments (non exhaustive) is listed below.

Specific comments:

Abstract: I suggest to focus the abstract (according to the comments above) having in mind the novelty and the scope of the manuscript.
Abstract: The abstract could benefit from a more quantitative description of the main results.

Line 1: The manuscript investigates rock temperatures, which have an impact on many topics also beyond rock wall instabilities (ecology, biology...). I suggest to extend the rationale to clarify the potential influence of the study.

Figure 2: Please show the location of the loggers on the images. The quality of the figure is not high, I guess this can be improved in the reviewed manuscript. Line 121: If there are any important overlapping methodological points with other papers it would be helpful to explain this more explicitly.

Figure 3 (and 5 later): It would be beneficial to show – maybe in the Appendix – the entire time series of the measurements (and of the modelling results for Fig. 5).

Line 161: what about rock joints? The bedrock is limestone – heavily fractured – as mentioned in the manuscript and shown in the figures.

Table 2: this could include the references directly in the table.

Line 226: is the sea temperature constant throughout the entire simulation for all the three scenarios?

Line 319 and Figure 6: what happens in summer? A short explanation would complete the paragraph and in case could also lead to an extension of the figure.

Line 476: this paragraph has no connection with the rest of the manuscript, I suggest to avoid it. Possibly it could be mentioned as an outlook.
* * *

---

## Author Comment (AC1) · 3 Mar 2021

**Response to Anonymous Referee #1**

We would like to thank the reviewer for the detailed and useful comments, which have helped to improve the quality and readability of our manuscript. In the following, we provide a reply to the points discussed by the reviewer as well as the changes in the manuscript.

The comments of the reviewer are written in **bold**, the extracts of the manuscript in *italics* with changes highlighted in *blue*.

**General comments:**

**This paper presents a four-year measurement series of near rock surface temperatures at coastal cliffs or canyon walls close to the Kongsfjord in the Ny Ålesund region, Svalbard and uses the well-known model CryoGrid 3 to calculate the energy balance at the observation sites and discusses the results. The authors show that the coastal sites are in general warmer than the sites a little bit further away from the fjord. This paper is carefully elaborated and shows interesting data and model results, which is an important contribution to a better understanding of the surface energy balance and related heat fluxes at a place where until now not many studies have been carried out in steep cliffs closely situated to a fjord. In general, before publishing this valuable article some major changes should be considered, which are outlined below:**

We appreciate that the reviewer identifies the relevance of our work and states it as an important contribution to a better understanding of thermal regimes in high Arctic rock walls. We understand the concerns of the reviewer with respect to neglecting the influence of snow, the assumption of a neutral atmospheric stratification and other points. We made changes to our manuscript including new model runs and clarifications in the text. Therefore, we are now confident that the revised manuscript is an improved version of our study and suitable for publication.

**Line 167, Neglecting influence of snow:**

**It is somehow understandable that the influence of snow is neglected for the investigated sites as the cliffs are steep and the influence of snow seems according to the authors until now not to be very important. However, there are two important points, which the authors do not consider enough in their paper, although they discuss the influence of snow in the paragraph of the discussing section on line 421 but this discussion cannot be used to justify their main approach of neglecting snow completely in their model approach.**

**- Firstly, recent studies show clearly that even in steep rock walls, small or even larger amounts of snow can accumulate on rock ledges and influence rock wall temperatures and change heat fluxes considerable (e.g. Haberkorn et al. 2015, 2016 and 2017) and their logger RW01 and their comment in the caption of table 5 shows that snow seems already today have some influence at certain cliff sites!**

**- Secondly, performing climate scenarios coupled to model runs of CryoGrid 3 not taking into account snow for future conditions is not at all reasonable as in the future probably snow cover is may going to**

**increase considerable during winter time particularly at sites which are close to a large moisture source (North Atlantic Ocean).**

**Therefore, I suggest to a) include in CryoGrid 3 a scenario including snow, which should not be a big problem as CryoGrid 3 is already well prepared and b) omit the future scenario runs in this paper (mainly chapter 4.5 and 5.4), as this part does not add any important additional information or only results, which are extremely speculative as the authors admit themselves on line 474. Particularly as the authors cannot really convincingly show and explain that snow is not going to play an important role in the future. However, a future snow cover at the small cliff sites can strongly change the whole thermal regime (e.g. the canyon site could be filled by more snow and corresponding snow drift by wind in the future).**

We agree with the reviewer that snow cover can play a significant role in the thermal regime of rock walls. To account for this, we included a discussion about snow cover with additional model runs and explanations. The model runs were performed by following the approach of Magnin et al. (2017) and taking the snowfall data of the AROME-Arctic dataset as an input. We present one simulation with maximal snow height (the snowfall accumulates completely in the rock wall) and another run with a maximum height of 20 cm snow cover.

The simulations result in an overestimation of snow cover in early winter and an underestimation in late spring when comparing to the snow-covered RW01 logger. This can be explained by the fact that snow cover in the rock walls is mainly due to snowdrift and a buildup from the foot of the rock wall, until it covers the temperature logger. The buildup of snow needs time so that the snow cover is first overestimated. In late spring, the snowdrift needs a long time to melt, so that the snow cover is then underestimated. Our model is able to represent snow cover, which results from snowfall, however, the complex buildup by snowdrift cannot be simulated adequately. Therefore, we decided to exclude snow cover in our main study and focus on generally snow-free sites, which is the case for the majority of the measurement sites. Yet, we agree with the reviewer that this a very important point to discuss and included the snow cover in a comprehensive supplement as well as a short discussion in the main manuscript.

Please find the changes in our manuscript below:

*Line 188-189: Further evaluations on the possible influence of a snow cover are given in the supplement, where we present model runs considering snow cover in the rock walls following the model approach of Magnin et al. (2017).*

*Line 332-334: Simulations with snowfall provided by the forcing data cannot represent the temporarily occurring snow cover in the rock walls adequately. A distinct overestimation of snow cover in early winter and a clear underestimation in late spring result in significant deviations from the measured data. Results of simulations including a snow cover are provided in the supplement.*

*Line 448-449: Model runs in the supplement show that the temporarily occurring snow cover in the rock walls poorly correlates with snowfall, but can mainly be traced back to snowdrift and in some cases the buildup of snow from the foot of the rock wall.*

A detailed discussion about snow cover is provided in a supplement to the manuscript:

*Supplement, Line 35-85:* **Supplement 2: Simulations considering snow cover**

*The measured RST data indicate temporarily occurring snow cover in some rock walls. As RW01 shows the most pronounced and regularly occurring dampening of the temperature signal, additional model runs were performed for this rock wall. Figure S2.1 displays a comparison of measured RST in RW01, a model run without snow cover (no snow scenario), a model run taking the maximum snowfall into account (snow scenario, max) and a model run restricting snow depth to a maximum of 20 cm (snow scenario, 20 cm) to account for the limited accumulation capacity of the steep rock wall. The simulations considering the snow cover were performed with CryoGrid 3, following the model approach of Magnin et al. (2017).*

*In January to mid of February, both snow scenarios overestimate the snow cover, leading to up to 10 °C higher RST, while the no snow scenario matches the measured data significantly better. This leads to the assumption that RW01 was not covered by snow in early winter despite the occurring snowfall. In March and April, both snow scenarios represent the measured data better than the no snow scenario. This is especially true for the snow scenario, max, which implies a thicker snow cover than expected for the vertical rock wall. In May and June, the measured data in RW01 still indicates the presence of a snow cover, while both snow scenarios underestimates the influence of snow in the rock wall. Consequently, the measurements suggest that the rock wall RW01 had a significantly thicker snow cover than calculated from the snowfall produced by the forcing data.*

*This leads to the conclusion that our model is able to represent snow cover and calculate a dampening of the temperature signal in general. However, the snowfall provided by the forcing data leads to high errors and cannot represent the conditions under snow cover in the rock wall. This can be explained by the fact that the snow cover in the rock wall does not directly correlate with the snowfall of the forcing data. Instead, it is likely that a snowdrift builds up from the foot of the rock wall in winter until it eventually covers the temperature logger. Therefore, both snow scenarios overestimate the snow cover in early winter, when the snow at the foot of the rock wall does not reach the temperature logger yet, and underestimates the snow cover in late spring and early summer, when the massive snow drift needs more time for melting than. As our model can only represent snow cover produced by the snowfall of the forcing data, but not the effect of snowdrift in complex terrain, we excluded periods with snow cover from the analysis in this study and focussed on measurement sites, which were found to be largely snow-free.*

[Figure]

*Figure S2.1: Comparison of measured daily RST in RW01 to a simulation without snow cover (no snow scenario), a model run taking the maximum snowfall into account (snow scenario, max) and a model run limiting snow depths to a maximum of 20 cm (snow scenario, 20 cm). The no snow scenario shows no dampening of the temperature signal, while both snow scenarios represent the snow cover well in late winter, but show an overestimation in early winter and an underestimation in late spring.*

*Future snow conditions will be characterized by a reduction of days with snow cover all over Svalbard. The snow water equivalent will be highly dependent on the emission scenario: Simulations of RCP4.5 show that snow water equivalent will be the same or slightly more, while simulations of RCP8.5 indicate a reduced maximum snow water equivalent of 50 % or more (Hanssen-Bauer et al., 2019). While the development of a snow cover in the analysed rock walls and its influence on RST is highly uncertain, we compare two future runs for RCP4.5 with no snow accumulation in the rock wall and snow accumulation to a maximum of 20 cm in the rock wall, respectively (Fig. S2.2). In the simulations, the snow cover reduces MARST by 0.25 °C to 1.21 °C. The lowering of MARST by the snow cover is due to the albedo increase in spring, which cools the surface by reflecting short-wave radiation. However, these results must be regarded as a rough approximation, as snowdrift and not snowfall seems to be the main driver of snow cover in these rock walls. As in most cases no significant snow cover could be detected in our measurement data, MARST might be closer to the scenario neglecting the snow cover, but might be reduced at sites where snowdrifts influence RST.*

[Figure]

*Figure S2.2: MARST of past and future simulations of the near-coastal scenario with RCP4.5 comparing two model runs neglecting the snow cover and taking potential snow cover up to 20 cm into account.*

**Line 121, assumption neutral atmospheric stratification:**

**It is understandable that the authors try to simplify the very complex processes of the turbulent fluxes. However, assuming that the conditions at the cliff walls could be approached by using for all cases neutral atmospheric stratification is not justified. This assumption would mean that an air parcel**

**moving close to the cliff would always have the same temperature (and density) as the surroundings at this position. This seems to be justified only for certain conditions during the year. The authors also justify in the discussion section their approach by tuning the roughness length until the model fits the observed values. This is reasonable to do. However, their estimated roughness length is, at least after my knowledge of such values and having seen the pictures of the measurement sites in the paper, at least one order of magnitude smaller what should be probably taken as roughness length for these specific observation sites.**

We agree that the assumption of a neutral atmospheric stratification cannot represent the conditions at the rock walls throughout the year. However, fitting the roughness length is a pragmatic approach as used by Magnin et al. (2017) and as the reviewer states. We estimated the roughness length to 0.018 m. As a general rule of thumb, the roughness length should be selected as 1/10 of the "roughness elements" of the surface. Given the strong variability of possible roughness elements (small-scale heterogeneity on the order of centimeters to decimeters, larger-scale roughness of protruding wall parts on the scale of meters), it is not possible to exactly determine a roughness length with this concept, and we resorted to fitting an "effective roughness length". However, we note that our chosen value is within the (admittedly large) range of values that are in agreement with the rule of thumb.

To clarify this point in our manuscript, we made the following changes:

*Line 200-204: An important fitting parameter was the roughness length $z_0$ as performed in Magnin et al. (2017). We set it to a value of 0.018 m, which represents roughly 1/10 of the height of the surface roughness elements. This fitted well to the* small-scale variations on the *rugged rock surface characterized by joint systems (Fig. 2)*, but uncertainties regarding the different spatial scale of roughness elements in the rock walls remain.

*Line 429-432: A critical point is the assumption of a neutral atmospheric stratification perpendicular to the vertical rock wall, which must be regarded as a first order approximation* as it does not account for the complex wind field and boundary layer conditions near the rock wall*. This leads to uncertainties in the near surface turbulent exchange of the vertical wall as micro-topography and changing weather conditions can influence the movement of air parcels.*

**Line 107, table 1 setting of surface temperature loggers:**

**The temperature loggers are located all at expositions of NE (except two of them ENE and N). It would be interesting to see the influence of the different expositions, which could be easily modelled by CryoGrid 3. I would assume that at this latitude the expositions do not play a very important role but it would be an interesting question which could be answered by CryoGrid 3.**

We thank the reviewer for this suggestion and included it in our revised manuscript. To do so, we ran simulations for north- and south-facing rock walls for the open fjord scenario and the non-coastal scenario. Both show variations up to 0.7 °C, which is substantially less than in lower latitudes like in the European Alps. In our manuscript, we changed the following:

*Line 327-331: The model results show differences in MARST according to the exposition of the rock wall: In the open fjord scenario, the lowest MARST in 2017/18 is found on the north-facing rock wall RW05 (-*

*0.9°C), while the highest MARST is calculated for RW06 facing east-north-east (-0.3 °C). Model runs for north- and south-facing rock walls suggest that differences in MARST due to exposition are only 0.7 °C or less. While no effect is detectable in winter, higher RST variations up to 1.6 °C are calculated for the spring season.*

*Line 442-447: Svalbard lies in the continuous permafrost zone (Brown et al., 1997; Obu et al., 2019) and deep permafrost is observed in the area, e.g. within the abandoned mine shafts (Liestøl, 1977) and in boreholes (Christiansen et al., 2010). We found that the exposition of the rock wall only leads to small differences in MARST. In the winter season, polar night conditions suppress any dependence on exposition, while the effect is most pronounced in spring season for low sun angles. In general, the influence of exposition in the high Arctic is small compared to sites at lower latitudes, like the European Alps (Gruber et al., 2004b; Magnin et al., 2015; Noetzli and Gruber, 2009).*

**Line 477, coastal cliffs in the high Arctic – a future geohazard:**

**This chapter does not add any important new information to the main topic of the paper. Please delete this section.**

We follow the suggestion of the reviewer and deleted the chapter 5.5. Instead, we included a significantly shorter paragraph in 5.4 to account for the main message of the deleted chapter.

*Line 516-522: Our model results indicate a significant warming of permafrost temperatures and a deepening of ALT in the 21$^{st}$ century, a trend that can lead to destabilization of rock slopes (Krautblatter et al., 2013). Besides, loss of sea ice and correlated longer duration of open-water season can enhance coastal erosion (Barnhart et al., 2014). In this study, the thermal regime of relatively low coastal cliffs is investigated. Indeed, similar processes can also affect much higher cliffs. Failures of coastal rock slopes can impact the water body and trigger displacement waves along shorelines, as happened in Paatuut / Greenland in 2000 (Dahl-Jensen et al., 2004; Hermanns et al., 2006). Due to permafrost degradation, rock slope failures in the high Arctic might become more likely in future, which should be taken into account for risk assessment of settlements and infrastructure.*

**Specific comments:**

**1. Line 31: better use: warming of atmosphere than warming of climate. The atmosphere can warm but the climate can only change but not warm.**

Fair enough! We have changed that sentence:

*Line 34: As a response to climate change, degradation of permafrost…*

**2. Line 46: may add the new literature from Etzelmüller et al. 2020; Etzelmuller, B., Guglielmin, M., Hauck, C., Hilbich, C., Hoelzle, M., Isaksen, K., Noetzli, J., Oliva, M., Ramos, M. (2020) Twenty years of European Mountain Permafrost Dynamics – the PACE Legacy. Environmental Research Letters 15, 14.**

We added the suggested literature in the manuscript.

*Line 48-50: The climatic changes are also apparent in permafrost temperatures on Svalbard as observed in boreholes over the last decades (Boike et al., 2018; Christiansen et al., 2010; Etzelmüller et al., 2020; Isaksen et al., 2007).*

**3. Line 50: May add some more literature here such as: Gisnås, K., Westermann, S., Schuler, T.V., Melvold, K., Etzelmüller, B. (2016) Small-scale variation of snow in a regional permafrost model. The Cryosphere 10, 1201-1215. Gisnås, K., Westermann, S., Schuler, T.V., Litherland, T., Isaksen, K., Boike, J., Etzelmüller, B. (2014) A statistical approach to represent small-scale variability of permafrost temperatures due to snow cover. The Cryosphere 8, 2063-2074. Haberkorn, A., Wever, N., Hoelzle, M., Phillips, M., Kenner, R., Bavay, M., Lehning, M. (2017) Distributed snow and rock temperature modelling in steep rock walls using Alpine3D. The Cryosphere 11, 585-607.**

We added the suggested literature in the manuscript.

*Line 55-56: …effectively modulated in presence of insulating snow cover (Gisnås et al., 2014, 2016; Haberkorn et al., 2015a, 2017).*

**4. Line 51: Exposition is not only important for steep rock walls. It is in general important also for less inclined slopes particularly at lower latitudes; already much older literature has shown this.**

We agree with the reviewer, that exposition plays as well an important role for less inclined slopes. To not exclude them, we changed the following sentence in our manuscript:

*Line 56-57: The terrain exposure induces significant spatial variability of short-wave radiation that should be considered when modelling thermal conditions in inclined slopes.*

**5. Line 82: please give a value for the altered net short-wave radiation through the decrease in reflection so that a comparison to the value given for the change in downward long-wave radiation can be done.**

The value for changes in downward long-wave radiation is given for the winter season to explain the rise in winter temperatures. For a better comparison, we cite now the value for changes in net long-wave radiation in winter season. Furthermore, we added the largest changes in net short-wave radiation, which occur in summer season due to the decrease in reflection. The following changes were made in the manuscript:

*Line 86-89: The winter warming is linked to a change in net long-wave radiation of +3.9 ± 3.9 W/m$^{-2}$ per decade (Maturilli et al., 2015). The net short-wave radiation is mainly altered in summer season by +12.0 ±*

*12.0 W/m⁻² per decade due to the* decrease in reflection caused by a reduced snow cover duration (Hop
and Wiencke, 2019; *Maturilli et al., 2015).*

**6. Line 88: Is there no mean annual precipitation available in Ny-Ålesund after 2000?**

We replaced the number with the mean precipitation of the years 2000-2019:

*Line 93-94: Measured mean annual precipitation in Ny-Ålesund in the period 2000-2019 was 484 mm*
*(Norwegian Meteorological Institute, 2021).*

**7. Line 101: Please give some information about temperature logger calibration.**

In the revised manuscript, we provide uncertainty estimates for the temperature measurements based
on duplicate loggers, which were installed in close vicinity of the main logger to mitigate the risk of loss
and failure. These backup loggers were generally placed within 10 cm of the main sensor in exactly the
same aspect, but in different cracks or different (e.g. wider/narrower/differently shaped) parts of the
same crack. These data were not used in the original manuscript, but for the revised manuscript we have
used them to evaluate the combined uncertainty of the sensor+logger system and the effect of the
placement in the walls. For all rock wall sites, the differences in MARST between adjacent temperature
loggers (i.e. between the backup and the main temperature time series) were found to be less than 0.1 °C
for annual averages, while seasonal RST showed differences of less than 0.2 °C between adjacent
temperature loggers.

The following changes were done in the manuscript:

*Line 115-120: The temperature sensors were placed in deep cracks in the rock wall so that both sides of*
*the iButton are in direct thermal contact with the rock surface and the sensor is protected from sunlight*
*(Fig. 2c). At each measurement site, we installed at least one more iButton, generally placed within 10 cm*
*of the main sensor in exactly the same aspect, but often in different cracks or different parts of the same*
*crack. We used these duplicate measurements to evaluate the combined uncertainty of the sensor/logger*
*system and the placement in the rock walls. For all sites, the differences between the two sensors were*
*found to be less than 0.1 °C for annual averages, while seasonal averages showed differences of less than*
*0.2 °C.*

**8. Line 104: the expression 'non-coastal rock walls' seems not very adequate chosen as this canyon cliffs**
**are only about 600 m from the coast. In my view a 'non-coastal rock wall' would be several kilometers**
**away from the fjord. Please change the wording.**

We understand the concern of the reviewer that the expression can lead to misunderstandings. Therefore,
we changed the wording in the entire manuscript from "non-coastal" to "near-coastal". An example would
be:

*Line 42-43: In this study, we will focus on rock surface temperatures in steep coastal and near-coastal cliffs at a high Arctic site close to Ny-Ålesund, Svalbard.*

**9. Line 114/115: how is the latent heat effect considered. Please explain or give at least a reference where the reader could get more information.**

The calculation of the latent heat effect is explained in detail in Westermann et al. (2016) and we have added the reference to the statement:

*Line 127-129: CryoGrid 3 calculates rock temperatures by solving the heat equation, uses the surface energy balance as an upper boundary condition, and considers latent heat effects depending on water content of the substrate as performed in Westermann et al. (2015).*

**10. Line 176/177: please give a source for this volumetric ice and mineral content percentages.**

Park et al. (2020) analysed rock samples of carbonate rocks close to Ny-Ålesund and found a porosity of circa 0.5 % for fresh rock samples without cracks or large pores. In our study, we are not looking at undisturbed material, but have to take fractures in the rock into account. Therefore, we increased the literature value up to 3 % for the volumetric water content to account for this. As the value of 3% is a coarse estimate, it was included in the sensitivity study in supplement 1. We added an explanation to the text and added the source in table 2:

*Line 194-199: We considered the bedrock to have a volumetric mineral content of 97 % and a volumetric water content of 3 %, which implied saturated conditions during the entire simulation. The assumed porosity was selected higher than measurements of 0.5 % of fresh carbonate samples without cracks in the Ny-Ålesund region (Park et al., 2020), with the goal to account for the fractured nature of the rock walls. Due to the high uncertainty of this value, a sensitivity study was performed for the volumetric mineral and water content.*

**11. Line 185, table 2: please give information about the source of your values you show in table 2. You could add a column in the table and show the references.**

We followed the suggestion of the reviewer and added a column in table 2 to show the references. Furthermore, we added the missing references in the text.

*Line 194-199: We considered the bedrock to have a volumetric mineral content of 97 % and a volumetric water content of 3 %, which implied saturated conditions during the entire simulation. The assumed porosity was selected higher than measurements of 0.5 % of fresh carbonate samples without cracks in the Ny-Ålesund region (Park et al., 2020), with the goal to account for the fractured nature of the rock walls. Due to the high uncertainty of this value, a sensitivity study was performed for the volumetric mineral and water content.*

*Line 204-209: We set the albedo for the horizontal ground surface to 0.15 (Westermann et al., 2009) and for water surfaces to 0.1, which is in the range of the surface ocean albedo for the typical high solar zenith angles in Svalbard (Li et al., 2006; Robertson et al., 2006). The albedo for ice and snow was set to a relatively low value of 0.55, as the highest influence of reflected short-wave radiation was expected for spring, when snowmelt decreases the albedo. This is in line with the reported decrease in albedo from 0.8 to 0.5 in Westermann et al. (2009). All values can be found in Table 2 and a sensitivity study for selected parameters is provided in the supplement.*

**Table 1: Model parameters assumed in the simulations.**

| Parameter | | Value | Unit | Reference |
|---|---|---|---|---|
| Albedo rock wall | $\alpha$ | 0.30 | [ - ] | *Blumthaler and Ambach (1988)* |
| Albedo ground | $\alpha_g$ | 0.15 | [ - ] | *Westermann et al. (2009)* |
| Albedo open water | $\alpha_w$ | 0.1 | [ - ] | *Li et al. (2006)* |
| Albedo melting snow / ice | $\alpha_s$ | 0.55 | [ - ] | *Westermann et al. (2009)* |
| Emissivity | $\varepsilon$ | 0.97 | [ - ] | *Bussieres (2002)* |
| Roughness length | $z_0$ | 0.018 | [m] | - |
| Mineral fraction | $mineral$ | 0.97 | [ - ] | *modified after Park et al. (2020)* |
| Water and ice fraction | $waterIce$ | 0.03 | [ - ] | *modified after Park et al. (2020)* |
| Water bucket depth | $d$ | 0.001 | [m] | - |

**12. Line 194/195: Is the effect (transition between fjord and land) you describe here really resolved? I can hardly believe this!**

The effect is not resolved, but as the grid cell covers both the fjord and land surface, it is the best possible solution to use it as forcing data for the shoreline. To avoid misunderstandings, we phrased the sentence differently:

*Line 220-222: The selected grid cell covers both parts of the fjord and the adjacent land surface and therefore provides suitable forcing data for the loggers located directly or within a short distance to the shoreline.*

**13. Line 203: why do you use the radiation data from AROME-Arctic dataset when you have much better data from the BSRN stations. Please clarify?**

We stayed with the data of AROME-Arctic throughout our study (using radiation, wind speed, rain/snowfall etc.) to use coherent forcing data and to avoid mixing data of different sources. The AROME-Arctic dataset was used as forcing data before also in other studies in the surroundings of Ny-Ålesund like in Zweigel et al. (2021). In our study, the only exception was air temperature to resolve local air temperature gradients.

Zweigel, R., Westermann, S., Nitzbon, J., Langer, M., Boike, J., Etzelmüller, B. and Schuler, T. V.: Simulating snow redistribution and ist effect on the ground thermal regime at a high-Arctic site on Svalbard, J. Geophys. Res.-Earth, in press, 2021.

**14. Line 240, table 4: RW01 is modeled according to table 4, but it is shown that this site is covered by snow in figure 3. Therefore, it is mandatory to include snow in the model scenarios for this logger otherwise you contradict yourself in the paper (see also the general comments about snow).**

As described in the reply to the comment "neglecting the influence of snow", the snow cover at the rock walls are mainly due to snowdrift and therefore, it is difficult to represent in our model. Consequently, we decided to focus our analysis on the periods without snow cover in the rock walls. To analyse the effect of snow, we added a supplement in our manuscript. The changes in the manuscript can be found under the comment "neglecting the influence of snow".

**15. Line 255, table 5: your model does not include snow but you wrote that only one logger is snow covered RW01 (figure 3). Please clarify this as it is very important for your assumption that there is no snow cover at the sites.**

We clarified in our manuscript which rock walls were affected by snow cover. To do so, we added a sentence in the paragraph of figure 3 and distinguished in table 5 between missing data and snow cover. While snow cover seems to occur at RW01 on a regular basis in spring and over a period of two to three months (in all years except for 2018 in our measurement period), the other rock walls are only affected occasionally and over a shorter time period: in May 2019 RW05 was affected, in May 2020 RW05, RW06 and RW08. We completely agree with the reviewer, that this information has to be given more clearly in our manuscript. The explanation why we excluded snow cover from our analysis is given in supplement 2.

We made the following changes in our manuscript:

*Line 184-189: We did not consider snow cover in the model, which was adequate for most of the measurement data in the analyzed time period from 2016 to 2020. An exception is displayed in Fig. 3, showing the damped signal of RW01 due to snow cover. Besides spring 2017, RW01 was influenced by snow cover over a period of two to three months in spring 2019 and 2020. In the other measurement sites, snow cover was only shortly observed in May 2019 (RW05) and in May 2020 (RW05, RW06, RW08). Further evaluations on the possible influence of a snow cover are given in the supplement, where we present model runs considering snow cover in the rock walls following the model approach of Magnin et al. (2017).*

*Table 5: Measured MARST and mean RST in the winter season (Dec-Feb) for all locations RW01 to RW08. Lack of data results from either (1) snow cover or (2) missing records. MARST are coldest in near-coastal settings (RW01 - RW03). Mean RST in winter season are found to be coldest in near-coastal settings, closely followed by settings in the bay (RW08), while stings at the open fjord show highest RST (RW04 – RW07).*

| Location | Site | Entire year | | | | Winter: Dec-Feb | | | |
|---|---|---|---|---|---|---|---|---|---|
| | | 2016/17 | 2017/18 | 2018/19 | 2019/20 | 2016/17 | 2017/18 | 2018/19 | 2019/20 |
| RW01 | Near-coastal | - [1] | -1.5 | - [1] | - [1] | -8.1 | -6.7 | -9.0 | -13.1 |
| RW02 | Near-coastal | -2.2 | -1.8 | -2.4 | -4.3 | -8.5 | -6.6 | -9.6 | -13.5 |
| RW03 | Near-coastal | -1.8 | -2.0 | -2.4 | -4.1 | -9.2 | -7.2 | -10.0 | -13.8 |
| RW04 | Open fjord | - [2] | -1.0 | -2.1 | -3.6 | - [2] | -5.1 | -8.7 | -12.0 |
| RW05 | Open fjord | -0.9 | -0.8 | - [1] | - [1] | -6.6 | -5.2 | -7.2 | -10.4 |
| RW06 | Open fjord | - [2] | -0.6 | - [2] | - [1] | - [2] | -5.0 | -8.5 | -11.5 |
| RW07 | Open fjord | - [2] | -0.8 | - [2] | -3.6 | - [2] | -4.6 | -7.6 | -11.1 |
| RW08 | Bay | -1.4 | -0.9 | -2.1 | - [1] | -8.6 | -5.8 | -9.7 | -13.1 |

**16. Line 275, figure 4: why is the variability (daily values?) not higher in comparison to figure 3 where there is much more variability in the same data. please clarify.**

Figure 3 does not show significantly higher variability in daily mean values. It might appear because the plot contains temperature data of more loggers and the time period is longer. Please find here a plot with similar time scales (both over 3 months) and the same loggers (RW02, RW04, RW08). RW04 is not included in the first plot, as it was not installed yet.

[Figure]

**17. Line 335, figure 7: What is Fub in the figures? Is this not G as noted in the figure caption?**

We thank the reviewer for this comment. The correct label in figure 7 is of course "G". We changed it in the revised manuscript:

[Figure]

*Figure 1: Surface energy balance for the seasons of the year for a vertical rock wall with an aspect of 40° (RW02, RW04 and RW08). Most pronounced differences in the scenarios are found in winter and spring, while summer and fall show similar fluxes. SW = net short-wave radiation; LW = net long-wave radiation; Qe = latent heat flux; Qh = sensible heat flux; G = ground heat flux.*

**Literature: Haberkorn, A., Wever, N., Hoelzle, M., Phillips, M., Kenner, R., Bavay, M., Lehning, M. (2017) Distributed snow and rock temperature modelling in steep rock walls using Alpine3D. The Cryosphere 11, 585-607. Haberkorn, A., Phillips, M., Kenner, R., Rhyner, H., Bavay, M., Galos, S.P., Hoelzle, M. (2016) Thermal regime of rock and its relation to snow cover in steep alpine rock walls: gemsstock, central swiss alps. Geografiska Annaler: Series A, Physical Geography 97, 579-597. Haberkorn, A., Hoelzle, M., Phillips, M., Kenner, R. (2015) Snow as a driving factor of rock surface temperatures in steep rough rock walls. Cold Regions Science and Technology 118, 64-75.**

The following literature was already included in our manuscript:

Haberkorn, A., Phillips, M., Kenner, R., Rhyner, H., Bavay, M., Galos, S. P. and Hoelzle, M.: Thermal regime of rock and its relation to snow cover in steep alpine rock walls: gemsstock, central swiss alps, Geografiska Annaler: Series A, Physical Geography, 97(3), 579–597, https://doi.org/10.1111/geoa.12101, 2015.

Haberkorn, A., Wever, N., Hoelzle, M., Phillips, M., Kenner, R., Bavay, M. and Lehning, M.: Distributed snow and rock temperature modelling in steep rock walls using Alpine3D, The Cryosphere, 11(1), 585–607, https://doi.org/10.5194/tc-11-585-2017, 2017.

In addition, we added the suggested literature

Haberkorn, A., Hoelzle, M., Phillips, M. and Kenner, R.: Snow as a driving factor of rock surface temperatures in steep rough rock walls, Cold Regions Science and Technology, 118, 64–75, https://doi.org/10.1016/j.coldregions.2015.06.013, 2015a.

in the introduction of our revised manuscript:

*Line 55-56: …effectively modulated in presence of insulating snow cover (Gisnås et al., 2014, 2016; Haberkorn et al., 2015a, 2017).*

---

## Author Comment (AC2) · 3 Mar 2021

**Response to Referee #2**

We would like to thank the reviewer for evaluating our manuscript and for the comments, which helped to improve it. We provide answers to the comments below as well as changes in the manuscript.

The comments of the reviewer are written in **bold**, the extracts of the manuscript in *italics* with changes highlighted in *blue*.

**General comments:**

**This manuscript presents a four-year time series of eight temperature loggers at rock cliffs in the surroundings of Ny-Ålesund. The authors use the model CryoGrid 3 in order to discuss the measurements and resolve the influence of the different components on the energy balance on the observations. In addition, the model is combined with three different representative concentration pathways in order to predict the evolution of rock cliff temperature and active layer thickness throughout the next century. The measurements advance our knowledge of the energy balance at rock cliffs in the Arctic, and the model is useful for their discussion. However, I have some concerns regarding the manuscript. In my opinion, the temperature measurements are not up to the state of art, and the modelling work is promising but could be largely integrated with more simulations: adding the two parts is still not sufficient for a publication.**

We appreciate that the reviewer acknowledges the advance in knowledge provided by our work. From his text we understand that his main concerns are the way the temperature measurements were performed, as well as the modelling work that could be enhanced by more simulations. To account for these points, we explain our sampling strategy and included clarifications, as well as new model runs in our revised manuscript. Please find the answers to the specific points below.

**Regarding the measurements: I could not find any information about the calibration of the temperature loggers. This is a major point of concern and strongly weakens all the sequent results and discussion. Additionally, I don't understand why the measurements have been performed with an accuracy of only 0.5 °C. In general, I would like to have an explanation of the sampling strategy, which is to my knowledge not up to the state of art in this field.**

We have experience with a wide range of measurement tools, including GeoPrecision (Magnin et al., 2019), Onset Hobo, Campbell and fully self-designed equipment. For this study in coastal cliffs of Ny-Ålesund, we chose to apply iButtons for several reasons:

1.  All installations of instruments in the surroundings of Ny-Ålesund have to follow the land owner's (KingsBay AS) regulations and land management plan, including minimal disturbance of the local environment. The coastal cliffs are under the influence of strong coastal erosion. Volumes of several cubic meters detach every year from localized spots, and smaller volumes have likely detached in the vicinity of our logger sites. If measurement equipment was taken down by such events, it would land directly on the tidal beaches, where seals, sea birds and reindeer regularly

rest and forage, and eventually enter the marine ecosystem. As iButtons are the most miniaturized and non-invasive tools, they clearly were the preferable method for this study.

2. Due to the rock type and the strong coastal erosion, parts of the rock walls are highly fractured, resulting in safety concerns when installing the temperature loggers. Placing iButtons in cracks takes minimal time and thus strongly reduces the exposure to risk, especially since natural rockfalls seem to occur mostly in spring, while we conducted our fieldwork in late summer. The "normal" equipment used to measure near-surface temperatures in rockwalls, for example in the European Alps, requires drilling holes to place the temperature sensor and mount the loggers. Vibrations from drilling would put additional stress on the rockwalls, thus increasing the risk for the operator, especially when installing the loggers from the foot of the wall. For this reason, we did not use equipment that requires drilling for this study, and will not do so in the future for these and similar sites.

Our measurements provide the very first temperature measurements of rock walls in the surroundings of Ny-Ålesund and to our knowledge in such a coastal high Arctic setting. We are of the opinion that a range of different systems are suitable to conduct measurements of rock near-surface temperatures, and that the most suitable system should be selected to fit the particular conditions of the study site. What ultimately matters is that the uncertainty of the measurements is low enough, so that the conclusions of the study can be secured in the light of this uncertainty. We have therefore made an effort to provide improved uncertainty calculations which not only relies on the manufacturer's accuracy of the loggers (point 1 below), but also takes the effect of the placement in the walls into account (point 2 below).

1. The manufacturer provides an accuracy of 0.5 °C for the iButtons. Similar to an ice-bath, this accuracy can be verified for the freezing point of water, if melting snow conditions can be identified in the time series, with temperatures remaining stable near 0 °C for longer periods. Due to the lack of a snow cover for most sites, we were not able not identify such episodes for most sites. However, we found one clear occasion where temperatures (RW01 in fig. 03) were found to be between -0.125 °C and 0.042 °C, which is well inside the given accuracy. This also corresponds to our experience from previous studies using iButtons, where we rarely detected iButtons, which featured an error of more than 0.25°C during melt conditions. For this reason, we did not calibrate the iButtons prior to deployment, as in previous work (e.g. Gisnås et al., 2014).

2. For each rockwall site, at least one backup iButton sensor was installed to mitigate the risk of equipment loss and failure (see above). These backup loggers were generally placed within 10 cm of the main sensor in exactly the same aspect, but in different cracks or different (e.g. wider/narrower/differently shaped) parts of the same crack. These data were not used in the original manuscript. For the revised manuscript, we have now used these duplicate measurements to evaluate the combined uncertainty of the sensor+logger system and the effect of the placement in the walls.

For all rock wall sites, the differences in MARST between the adjacent temperature loggers were found to be less than 0.1 °C for annual averages, while we base our conclusions on differences in MARST between different rock wall sites on the order of 1.0 °C (table 5). Furthermore, seasonal RST showed differences of less than 0.2 °C between adjacent temperature loggers. For winter season RST, we base our conclusions on temperature differences between different rock wall sites of 1.5 °C to 2.2 °C (table 5). We therefore conclude that our conclusions relating to the in-situ measurements are well supported in the light of the measurement uncertainty.

References:

Magnin, F., Etzelmüller, B., Westermann, S., Isaksen, K., Hilger, P., & Hermanns, R. L. (2019): Permafrost distribution in steep rock slopes in Norway: measurements, statistical modelling and implications for geomorphological processes. Earth Surface Dynamics, 7(4), 1019-1040.
Gisnås, K., Westermann, S., Schuler, T. V., Litherland, T., Isaksen, K., Boike, J., & Etzelmüller, B. (2014): A statistical approach to represent small-scale variability of permafrost temperatures due to snow cover. The Cryosphere, 8(6), 2063-2074.

The following changes were done in the manuscript:

*Line 115-120: The temperature sensors were placed in deep cracks in the rock wall so that both sides of the iButton are in direct thermal contact with the rock surface and the sensor is protected from sunlight (Fig. 2c). At each measurement site, we installed at least one more iButton, generally placed within 10 cm of the main sensor in exactly the same aspect, but often in different cracks or different parts of the same crack. We used these duplicate measurements to evaluate the combined uncertainty of the sensor/logger system and the placement in the rock walls. For all sites, the differences between the two sensors were found to be less than 0.1 °C for annual averages, while seasonal averages showed differences of less than 0.2 °C.*

**Regarding the modelling: a sensitivity study to the many model parameters would be beneficial to the conclusions of the paper and could, with a proper set up, provide interesting insights in the investigated processes. In the modeling in general, and in particular for the future scenarios, the quantification of the uncertainty (related to the climate scenarios and their propagation in the modelling) is required.**

We agree with the reviewer that the quality of our manuscript can benefit from a sensitivity study. We performed five additional sets of simulations, varying the parameters roughness length, mineral fraction, albedo of the rock surface, exposition of the slope as well as increasing seawater temperature for the future simulation. We included this analysis as a supplement in our manuscript:

*Supplement, Line 1-33: **Supplement 1: Sensitivity analysis to model parameters***

*To analyse the sensitivity of our model results to different input parameters, we performed five additional sets of simulations. We varied the parameters roughness length, mineral fraction (which is equal to one minus porosity) and albedo of the rock surface for which we could not obtain reliable values. Furthermore, we analysed the parameter exposition as it can change in the rock wall because of small ledges and corners at the surface. To account for uncertainties in changes of seawater temperature, we performed model runs with an increase of 1.0 °C and 2.0 °C in seawater temperature until 2100 for the RCP4.5 scenario.*

*We found the strongest deviations from the reference simulation when varying the roughness length. Changes in the range of a few millimetre (± 0.005 m) result in only small deviations in modelled MARST (< 0.1 °C), while an increase in the range of decimetres (+ 0.1 m) can lead to over 0.5 °C lower MARST. These findings emphasize that the roughness length is a crucial factor for the calculation of RST, highlighting the role as a fitting parameter.*

*When varying the mineral fraction up to 0.1, the modelled MARST changed insignificantly (< 0.1 °C), suggesting that the calculation of RST is robust against variations in the mineral fraction. This finding is*

*especially important as the exact value of the mineral fraction for the rock walls was not known in this study.*

*Varying the exposition of the slope up to 10° had almost no effect on the modelled MARST (< 0.1 °C). The insignificant deviations can be explained by the high latitude of the field site and the prevalent polar night and polar day conditions.*

*We also analysed the sensitivity of the model results against changes of the rock surface albedo, as the albedo of carbonates can vary in different field conditions and the exact value was not known. Changes up to 0.02 did not substantially affect the modelled MARST (< 0.1 °C). However, reducing the rock surface albedo by 0.1 resulted in clearly lower modelled MARST (up to 0.26 °C).*

*Future simulations of the RCP4.5 scenario show an increase in modelled MARST of up to 0.1 °C and 0.14 °C for an increase in seawater temperature until 2100 by 1.0 °C and 2.0 °C, respectively. Hanssen-Bauer et al. (2019) states that the surface waters around Svalbard will increase by 1.0 °C in 50 years from now in the RCP4.5 scenario, but regional deviations are likely. However, the sensitivity analysis shows that MARST of the coastal cliffs in our field site will only be affected to a slight extent.*

*To conclude, the results of the sensitivity analysis found that the parameters were in most cases robust against variations, with  roughness length and rock surface albedo being the most sensitive parameters.*

**Due to the limits of the temperature time series and the current state of the modelling, I suggest to restructure the manuscript in order to provide a more thorough study. Personally, I suggest to focus the manuscript on the modelling part: use the observations for model calibration and then use this to perform a more complete series of synthetic experiments to investigate the energy balance in different conditions. Therefore, I consider the manuscript promising and potentially suited for publication, but I suggest some major revisions prior to publication. A short list of specific comments (not exhaustive) is listed below.**

We have provided a thorough assessment of the uncertainty of our measurements, which shows that  our conclusions are well supported by our measurements. In particular, they convincingly show the influence of sea ice cover, which is a key finding of our study. As the in-situ data set from such a high-latitude location is unique, we would like to keep the measurements as a prominent part of our manuscript.

We followed the suggestion of the reviewer to perform additional model runs with several sensitivity studies that we included in the supplement. Hereby, we set our focus on the parameters roughness length, porosity and albedo of the rock surface, for which the values were poorly constrained. Furthermore, we analysed the influence of varying exposition of the slope and increasing seawater temperature. All results can be found in the supplement.

**Specific comments:**

**Abstract: I suggest to focus the abstract (according to the comments above) having in mind the novelty and the scope of the manuscript.**

**Abstract: The abstract could benefit from a more quantitative description of the main results.**

Following the comment of the reviewer, we highlight the novelty of the manuscript in the abstract and point out that these are one of the first measurements of RST in steep rock walls in the high Arctic. We made the following changes in our manuscript:

*Line 16-17: This study presents one of the first comprehensive datasets of rock surface temperature measurements of steep rock walls in the high Arctic, comparing coastal and near-coastal settings.*

Furthermore, we agree with the reviewer that the manuscript could benefit from a more quantitative description of the main results. Therefore, we made the following changes in the manuscript:

*Line 19-26: Our measurements comprise four years of rock surface temperature data from summer 2016 to summer 2020. Mean annual rock surface temperatures ranged from -0.6 °C in a coastal rock wall in 2017/18 to -4.3 °C in a near-coastal rock wall in 2019/20. Our measurements and model results indicate that rock surface temperatures at coastal cliffs are up to 1.5 °C higher than near-coastal rock walls when the fjord is ice-free in winter season, resulting from additional energy input due to higher air temperatures at the coast and radiative warming by relatively warm seawater. An ice layer on the fjord counteracts this effect, leading to similar rock surface temperatures as in near-coastal settings. Our results include a simulated surface energy balance with short-wave radiation as the dominant energy source during spring and winter with net average seasonal values of up to 100 W/m², and long-wave radiation being the main energy loss with net seasonal averages between 16 W/m² and 39 W/m².*

**Line 1: The manuscript investigates rock temperatures, which have an impact on many topics also beyond rock wall instabilities (ecology, biology…). I suggest to extent the rationale to clarify the potential influence of the study.**

We added in the introduction that degradation of mountain permafrost can impact the local ecology and that the thermal state of the ground is also an important parameter for landscape development. Furthermore, we included the impact on coastal erosion and local ecology, which is important at our field site due to breeding seabirds. We made the following changes in our manuscript:

*Line 34-35: As a response to climate change, degradation of mountain permafrost can impact local ecology (Jin et al., 2020), play an important role in landscape development (Etzelmüller and Frauenfelder, 2009) and contribute to slope destabilization…*

*Line 40-42: However, permafrost dynamics in steep rock walls in the high Arctic are poorly understood, despite the impact on coastal erosion (Ødegård and Sollid, 1993) and local ecology such as breeding seabirds (Yuan et al., 2010).*

**Figure 2: Please show the location of the loggers on the images. The quality of the figure is not high, I guess this can be improved in the revised manuscript.**

We followed the suggestion of the reviewer and marked the location of the loggers in the images. Furthermore, we changed the resolution of the images to a higher quality and exported it to a resolution of 330 dpi following the guidelines of The Cryosphere.

You can find the following changes in the manuscript:

[Figure]

*Figure 1: Locations of rock wall loggers used in this study: (a) coastal cliffs at the open fjord next to Ny-Ålesund airport (tidal zone visible in bottom). The position of RW06 is marked with a red circle; (b) near-coastal rock walls in the canyon of Bayelva. The position of RW01 is marked with a red circle; (c) close-up of a rock wall logger location: marking tape is visible, while the logger is located about 5 cm inside the crack in thermal contact with the rock.*

**Line 121: If there are any important overlapping methodological points with other papers it would be helpful to explain this more explicitly.**

We understand that more clarification is needed in this paragraph. Therefore, we made the following changes in the manuscript:

*Line 134-137: As a consequence, movement of air parcels at a vertical wall would be parallel to the surface rather than perpendicular. Therefore, we assumed in all model calculations, that the near-surface wind profile follow a neutral atmospheric stratification. To do so, we applied the same approach as performed by Magnin et al. (2017), who used CryoGrid 3 to simulate rock wall and permafrost temperatures at the Aiguille de Midi, France.*

**Figure 3 (and 5 later): It would be beneficial to show – maybe in the Appendix – the entire time series of the measurements (and of the modelling results for Fig. 5).**

We included a figure in the supplement to the manuscript, showing the entire time series:

[Figure]

*Supplement, Line 89-92: **Supplement 3: Time series of measurement data and model results***

*Figure S3.1: Time series of measurement data and model results ranging from 27.08.2016 to 31.08.2020 and including all rock walls RW01 to RW08.*

**Line 161: what about rock joints? The bedrock is limestone – heavily fractured – as mentioned in the manuscript and shown in the figures.**

The latent heat fluxes are determined by the water which is available for evaporation. Rock joints might act as pathways for the water, but do not play an important role in storing water close to the surface, where it can be evaporated. Therefore, the latent heat fluxes are mainly driven by evaporation, when the rock surface is wet from rainwater and a water bucket approach can sufficiently represent this process.

However, the influence of rock joints cannot be neglected completely. Therefore, we adapted the volumetric water content in our model and assumed a higher value than the literature gives for fresh rock samples without cracks. As this might lead to uncertainties in determining the volumetric water content, we included this parameter in our sensitivity study. The results are robust against deviations in the volumetric water content.

We made the following changes in the manuscript:

*Line 194-199: We considered the bedrock to have a volumetric mineral content of 97 % and a volumetric water content of 3 %, which implied saturated conditions during the entire simulation. The assumed porosity was selected higher than measurements of 0.5 % of fresh carbonate samples without cracks in the Ny-Ålesund region (Park et al., 2020), with the goal to account for the fractured nature of the rock walls. Due to the high uncertainty of this value, a sensitivity study was performed for the volumetric mineral and water content.*

*Supplement, line 14-16: When varying the mineral fraction up to 0.1, the modelled MARST changed insignificantly (< 0.1 °C), suggesting that the calculation of RST is robust against variations in the mineral fraction. This finding is especially important as the exact value of the mineral fraction for the rock walls was not known in this study.*

**Table 2: this could include the references directly in the table.**

We thank the reviewer for this suggestion and added a column in table 2 to show the references. Furthermore, we added the references in the text.

*Line 194-199: We considered the bedrock to have a volumetric mineral content of 97 % and a volumetric water content of 3 %, which implied saturated conditions during the entire simulation. The assumed porosity was selected higher than measurements of 0.5 % of fresh carbonate samples without cracks in the Ny-Ålesund region (Park et al., 2020), with the goal to account for the fractured nature of the rock walls. Due to the high uncertainty of this value, a sensitivity study was performed for the volumetric mineral and water content.*

*Line 204-209: We set the albedo for the horizontal ground surface to 0.15 (Westermann et al., 2009) and for water surfaces to 0.1, which is in the range of the surface ocean albedo for the typical high solar zenith angles in Svalbard (Li et al., 2006; Robertson et al., 2006). The albedo for ice and snow was set to a relatively low value of 0.55, as the highest influence of reflected short-wave radiation was expected for spring, when snowmelt decreases the albedo. This is in line with the reported decrease in albedo from 0.8 to 0.5 in Westermann et al. (2009). All values can be found in Table 2 and a sensitivity study for selected parameters is provided in the supplement.*

*Table 1: Model parameters assumed in the simulations.*

| Parameter | | Value | Unit | Reference |
|---|---|---|---|---|
| Albedo rock wall | $\alpha$ | 0.30 | [ - ] | Blumthaler and Ambach (1988) |
| Albedo ground | $\alpha_g$ | 0.15 | [ - ] | Westermann et al. (2009) |
| Albedo open water | $\alpha_w$ | 0.1 | [ - ] | Li et al. (2006) |
| Albedo melting snow / ice | $\alpha_s$ | 0.55 | [ - ] | Westermann et al. (2009) |
| Emissivity | $\varepsilon$ | 0.97 | [ - ] | Bussieres (2002) |
| Roughness length | $z_0$ | 0.018 | [m] | - |
| Mineral fraction | $mineral$ | 0.97 | [ - ] | modified after Park et al. (2020) |
| Water and ice fraction | $waterIce$ | 0.03 | [ - ] | modified after Park et al. (2020) |
| Water bucket depth | $d$ | 0.001 | [m] | - |

**Line 226: is the sea temperature constant throughout the entire simulation for all three scenarios?**

Hanssen-Bauer et al. (2019) states that the surface waters around Svalbard will increase by 1 °C in fifty years from now for the RCP4.5 scenario. However, stronger warming is expected in areas further south and other areas might even cool about 1 °C (Hanssen-Bauer et al., 2019). Therefore, it is difficult to assess the expected sea temperature increase in Kongsfjorden and the sea temperature is constant for all three scenarios.

Nevertheless, we want to account for the comment of the reviewer. We ran additional simulations for the RCP4.5 scenario with a sea temperature increase of 1 °C and 2 °C until 2100, respectively. Slight differences in MARST of up to 0.14 °C are expected at the end of the century.

We included the findings in the supplement:

*Supplement, Line 5-7: To account for uncertainties in changes of seawater temperature, we performed model runs with an increase of 1.0 °C and 2.0 °C in seawater temperature until 2100 for the RCP4.5 scenario.*

*Supplement, Line 26-30: Future simulations of the RCP4.5 scenario show an increase in modelled MARST of up to 0.1 °C and 0.14 °C for an increase in seawater temperature until 2100 by 1.0 °C and 2.0 °C, respectively. Hanssen-Bauer et al. (2019) states that the surface waters around Svalbard will increase by 1.0 °C in 50 years from now in the RCP4.5 scenario, but regional deviations are likely. However, the sensitivity study shows that MARST of the coastal cliffs in our field site will only be affected to a slight extent.*

**Line 319 and Figure 6: what happens in summer? A short explanation would complete the paragraph and in case could also lead to an extension of the figure.**

This paragraph analyzes the factors leading to different rock surface temperatures when sea ice is present. During our measurement period from 2016 – 2020, sea ice appeared only in the months given in figure 6, a comparison between conditions with sea ice and without sea ice is not possible for summer months.

To avoid possible misunderstanding, we made the following changes in the manuscript:

*Line 349-352: Between December and February, air temperature and the lack of radiative heating are the dominant factors, while reflected short-wave radiation plays no role. In March and April, the influence of reflected short-wave radiation increases as polar night conditions end. As no sea ice occurred after April in the measurement period 2016 to 2020, no analysis could be performed for late spring and early summer season.*

**Line 476: this paragraph has no connection with the rest of the manuscript, I suggest to avoid it. Possibly it could be mentioned as an outlook.**

We follow the suggestion of the reviewer and deleted the chapter 5.5. Instead, we included a significantly shorter paragraph in 5.4 to account for the main message of the deleted chapter.

*Line 516-522: Our model results indicate a significant warming of permafrost temperatures and a deepening of ALT in the 21$^{st}$ century, a trend that can lead to destabilization of rock slopes (Krautblatter et al., 2013). Besides, loss of sea ice and correlated longer duration of open-water season can enhance coastal erosion (Barnhart et al., 2014). In this study, the thermal regime of relatively low coastal cliffs is investigated. Indeed, similar processes can also affect much higher cliffs. Failures of coastal rock slopes can impact the water body and trigger displacement waves along shorelines, as happened in Paatuut / Greenland in 2000 (Dahl-Jensen et al., 2004; Hermanns et al., 2006). Due to permafrost degradation, rock slope failures in the high Arctic might become more likely in future, which should be taken into account for risk assessment of settlements and infrastructure.*

---

## Referee Report (RR1)

**Surface temperatures and their influence on the permafrost thermal regime in high Arctic rock walls on Svalbard" by Juditha Undine Schmidt et al.**

**Cicoira Alessandro (Referee)**
alessandro.cicoira@epfl.ch

This manuscript presents a four-year time series of eight temperature loggers at rock cliffs in the surroundings of Ny- Ålesund. The authors use the model CryoGrid 3 in order to discuss the measurements and resolve the influence of the different components on the energy balance on the observations. In addition, the model is combined with three different representative concentration pathways in order to predict the evolution of rock cliffs temperature and active layer thickness throughout the next century.

The measurements advance our knowledge of the energy balance at rock cliffs in the Arctic, and the model is useful for their discussion. However, I have some concerns regarding the manuscript. In my opinion, the temperature measurements are not up to the state of the art, and the modelling work is promising but could be largely integrated with more simulations: adding the two parts is still not sufficient for a publication.

**Answer to the author response:**

I provide two main answer to the author response. One with regard to the measurements, the other one regarding the structure of the paper and the modelling study. The other minor comments have been implemented and I don't need to add on them. In general, I consider my comments answered and the manuscript suitable for publication.

Regarding the measurements: iButtons are, despite great limitations, a valuable tool for field studies in difficult environments and I have myself used many during the years. When I mention the "state of the art in the field" I was referring already to the use of iButtons. So, I am not commenting the use of other technologies. In my experience with the iButtons, more than a logger is needed for each location (ideally three to have a standard deviation), calibration is essential (with a minimum of two points, possibly using a cold bath with mixing devices for the 0°C) and the accuracy of the instrument can be set up (according to the model between 0.125 and 0.5 or between 0.06 and 0.5 °C). Having quickly stated what the standard would be for me, I understand that in other environments and geographical settings constrains of every sort can limit the possibilities during programming and deployment. Also, I understand that the measurements have been done already and cannot be modified anymore: this is why I suggest not to change the sampling strategy but to explain it in more detail. This has been done in the author answer to my comments, and partially in the revised manuscript. Personally I find this point essential and suggest to extend the discussion on this point in the manuscript.

Regarding the modelling: my comments have been integrated mostly in the appendix with some small mentions in the text of the manuscript. In particular, my major review has been accommodated by the sensitivity study, while my (personal) suggestion to reshape the study has been correctly argued against and the original structure and idea of the paper has been maintained – I believe improved from the reviews.

With these two short comments I suggest the paper for publication in TC.

Best Regards,
Alessandro Cicoira